# Epistasis and entrenchment of drug resistance in HIV-1 subtype B

**Avik Biswas[1,2], Allan Haldane[1,2], Eddy Arnold[3,4], Ronald M Levy[1,2,5]\***

[1]Center for Biophysics and Computational Biology, Temple University, Philadelphia, United States; [2]Department of Physics, Temple University, Philadelphia, United States; [3]Center for Advanced Biotechnology and Medicine, Rutgers University, Piscataway, United States; [4]Department of Chemistry and Chemical Biology, Rutgers University, Piscataway, United States; [5]Department of Chemistry, Temple University, Philadelphia, United States

**Abstract** The development of drug resistance in HIV is the result of primary mutations whose effects on viral fitness depend on the entire genetic background, a phenomenon called 'epistasis'. Based on protein sequences derived from drug-experienced patients in the Stanford HIV database, we use a co-evolutionary (Potts) Hamiltonian model to provide direct confirmation of epistasis involving many simultaneous mutations. Building on earlier work, we show that primary mutations leading to drug resistance can become highly favored (or entrenched) by the complex mutation patterns arising in response to drug therapy despite being disfavored in the wild-type background, and provide the first confirmation of entrenchment for all three drug-target proteins: protease, reverse transcriptase, and integrase; a comparative analysis reveals that NNRTI-induced mutations behave differently from the others. We further show that the likelihood of resistance mutations can vary widely in patient populations, and from the population average compared to specific molecular clones.

DOI: https://doi.org/10.7554/eLife.50524.001

**\*For correspondence:** ronlevy@temple.edu

**Competing interests:** The authors declare that no competing interests exist.

## Introduction

HIV mutates rapidly as it jumps from host to host, acquiring resistance to each host's distinct immune response and applied drug regimen. Drug-resistance mutations (DRMs) arise when the virus evolves under selective pressure due to antiretroviral therapy (ART). Primary DRMs often incur a fitness penalty which is then compensated for by accompanying associated mutations (*Heeney et al., 2006; Shafer and Schapiro, 2008*). With the use of current robust inhibitors in drug therapy, the drug-resistance mutation patterns in HIV have become increasingly more complex (*Richman et al., 2004a; Iyidogan and Anderson, 2014*) often leading to ART failure in patients. Resistance is estimated to develop in up to 50% of patients undergoing monotherapy (*Richman et al., 2004b*) and up to 30% of patients receiving current combination antiretroviral therapy (c-ART) (*Gupta et al., 2008*). The primary drug targets in treatment of HIV are the enzymes coded by the *pol* gene, reverse transcriptase (RT), protease (PR), and integrase (IN). A large number of sequences of HIV are available for RT, PR, and IN for patients who have been treated during the past nearly 30 years, and this information permits critical sequence-based informatic analysis of drug resistance.

The selective pressure of drug therapy modulates patterns of correlated mutations at residue positions which are both near and distal from the active site (*Chang and Torbett, 2011; Haq et al., 2012; Flynn et al., 2015; Yilmaz and Schiffer, 2017*). A mutation's impact on the stability or fitness of a protein however is dependent on the entire genetic background in which it occurs: a phenomenon known as 'epistasis'. Drug resistance develops as these mutations accumulate, providing the virus a fitness benefit in the presence of drug pressure, with a complex interplay in the roles of

primary and secondary mutations (*Yilmaz and Schiffer, 2017*; *Ragland et al., 2017*). When a primary resistance mutation is incurred in the context of a wild-type background, there is usually a fitness penalty associated with it. In backgrounds with more (accessory) mutations however, the fitness penalty decreases and on average, the primary mutation can become more likely than the wild-type residue. Because the beneficial effects of the associated mutations depend on the primary mutation, with the accumulation of (accessory) mutations, the reversion of the primary mutation can become increasingly deleterious, leading to a type of evolutionary 'entrenchment' of the primary mutation (*Pollock et al., 2012*; *Shah et al., 2015*; *McCandlish et al., 2016*). The entrenchment effect on a primary mutation can be very strong on average, and is in fact, modulated by the collective effect of the entire sequence background.

The effective modeling of epistasis is then critical to the identification and understanding of the drug and immune pressure mediated mutational combinations that give rise to drug-resistant, stable viruses. Experimental techniques to assess the effect of multiple mutations on phenotype have proven effective (*Troyer et al., 2009*; *da Silva et al., 2010*; *Liu et al., 2013*), but functional assays to test all possible combinations are impossible because of the vast size of the mutational space. Co-evolutionary information derived from multiple sequence alignments (MSAs) of related protein sequences have also served as a basis for building models for protein structure and fitness (*Göbel et al., 1994*; *Lockless and Ranganathan, 1999*; *Morcos et al., 2011*; *Hinkley et al., 2011*; *Haq et al., 2012*; *Ferguson et al., 2013*; *Mann et al., 2014*; *Jacquin et al., 2016*; *Hopf et al., 2017*; *Tubiana et al., 2019*). A subset of such models, called Potts statistical models (*Levy et al., 2017*) (a generalization of *Ising spin glass* models), have been used successfully to predict the tertiary and quaternary structure of proteins (*Morcos et al., 2011*; *Marks et al., 2012*; *Sułkowska et al., 2012*; *Sutto et al., 2015*; *Haldane et al., 2016*; *Levy et al., 2017*; *Sjodt et al., 2018*) as well as viral protein stability and fitness (*Shekhar et al., 2013*; *Barton et al., 2016a*; *Hopf et al., 2017*; *Flynn et al., 2017*; *Levy et al., 2017*; *Louie et al., 2018*).

Recent studies using Potts models inferred the epistatic interactions involved in the evolution of drug resistance in HIV-1 protease (*Flynn et al., 2017*). The authors identified sequence backgrounds with differing patterns of associated mutations which were predicted to either strongly entrench or strongly disfavor particular primary mutations in HIV-1 protease. Of crucial importance is the observation that whether a mutation favors or disfavors a primary resistance mutation with which it is associated, depends on the entire sequence background. Here we confirm the predictions in *Flynn et al. (2017)* providing the first direct confirmations of 'entrenchment' by the entire sequence background and show how such models capture the epistatic interactions that lead to drug resistance in each of the three predominant target enzymes in HIV subjected to selective pressure of ART: protease, reverse transcriptase, and integrase. We show that it is possible to predict which mutations a particular background will support in the drug-experienced population and how conducive that background is towards that mutation. Understanding the relationship between the likelihood of a drug-resistance mutation and the specific sequence background is important for the interpretation of mutagenesis experiments, which are carried out in the background of specific reference sequences such as subtype B molecular clones NL4-3, HXB2, and others (*Martinez-Picado et al., 1999*; *An et al., 1999*). Our results demonstrate that the background-dependent likelihood of a primary mutation and the strength of compensatory behavior can vary widely in patient populations, and from population averages to predominant subtype B molecular clones such as NL4-3 or HXB2.

It is useful to review some basic properties of Potts models in this context. A Potts model is a maximum-entropy model of the likelihood of a dataset multiple sequence alignment (MSA), constrained to predict the bivariate (pairwise) residue frequencies between all pairs of positions in the alignment, in other words it is designed to capture the observed pairwise mutational correlations caused by epistasis. A central quantity known as the 'statistical' energy of a sequence $S$ (see *Equation 1*, Materials and methods) is commonly interpreted to be proportional to fitness; the model predicts that sequences will appear in the dataset with probability $P(S) \propto e^{-E(S)}$, such that sequences with favorable statistical energies are more prevalent in the MSA. A key feature of the Potts model is that the effect of a mutation on fitness and $E(S)$ is 'background-dependent', as a single change at one position causes a difference in $L - 1$ of the coupling values in the sum for $E(S)$, and a mutation at one position will affect mutations at all other positions both directly and indirectly through chains of interactions involving one or many intermediate residues. Because of this, the Potts model

predicts complex patterns of mutational correlations, even though the effective energy model includes only pairwise terms.

## Results

### Epistasis: the effect of the background on primary resistance mutations in HIV

We begin by demonstrating the importance of the genetic background in determining the frequencies of primary mutations in HIV and verifying this by comparison to the empirical background dependence of primary mutations observed in the Stanford HIV database. We do this using a background-dependent computation of residue biases. If we mutate a sequence $S$ at position $i$ to a residue $\alpha$, calling the mutated sequence $S^i_\alpha$, we can also calculate the mutant sequence's probability $P(S^i_\alpha)$ according to the Potts model. Further, we can calculate the relative probability of any mutation at position $i$ of sequence $S$, as $\mathbb{P}(S, i, \alpha) = P(S^i_\alpha)/\sum_\beta P(S^i_\beta)$. Notably, $\mathbb{P}(S, i, \alpha)$ depends on the background comprised of all positions except $i$. We will call this background $S \backslash i$ representing the sequence $S$ with an unspecified residue at position $i$. Given a very large sample of sequences such that the background $S \backslash i$ is sampled multiple times, a residue $\alpha$ will appear at position $i$ in that background with frequency $\mathbb{P}(S, i, \alpha)$. In practice, obtaining such large samples is difficult. But equivalently, if a background $S \backslash i$ is sampled from nature without knowledge of the residue at position $i$, the Potts model can be used to predict that the residue $\alpha$ will be at the missing position with a likelihood $\mathbb{P}(S, i, \alpha)$. We emphasize that the model was not fit using this information. While the Potts Hamiltonian model only contains fields and pairwise coupling terms; $\mathbb{P}(S, i, \alpha)$ involves combinations of couplings of residues at $i$ with all other positions and cannot be expressed as a simple function of the correlations between mutations at pairs of positions.

Using $\mathbb{P}(S, i, \alpha)$ we can classify sequences by how likely a residue $\alpha$ is to appear at a position $i$ (*Figure 1A*). We divide the dataset MSA into 10 groups of sequences ordered by $\mathbb{P}(S, i, \alpha)$, and compare the Potts predicted frequency of the residue in each group to its observed frequency. *Figure 1B* shows good agreement for the likelihood of M at position 90 in HIV-1 protease, an important drug-resistance mutation. *Figure 1D* shows that the average absolute error between Potts predicted and observed frequencies is very small for all major drug-resistance mutations in HIV PR, RT and IN, establishing that the Potts model is a good predictor of epistasis in viral sequence backgrounds. These results provide one of the most direct verifications of the complex patterns (involving the effect of many mutations) of epistatic interactions involving drug-resistance mutations in HIV. This result also shows that the background dependence can be very strong, as many sequences can have likelihoods of $\mathbb{P}(S, i, \alpha) > 0.9$ while others have $\mathbb{P}(S, i, \alpha) < 0.1$ as is seen for the DRM L90M in HIV PR (*Figure 1B*).

The epistatic predictions of the Potts model cannot be simply explained by the number of background mutations (or Hamming distance) relative to the wild-type (consensus wild-type subtype B amino acid sequence). It is not simply the number of mutations that bias residues, but particular patterns of mutations. The inset figure of *Figure 1B* shows that there is significant overlap in the distributions of the number of mutations between two groups of sequences in which the probability of observing the mutation L90M is ~3x smaller in one than the other ('blue' compared to 'red') illustrating that a prediction of residue frequencies based on Hamming distance alone is insufficient. We compare the residue-prediction ability of the Potts model using $\mathbb{P}(S, i, \alpha)$ to that of a 'Hamming' model which predicts residue frequencies purely based on the number of mutations. This is plotted in *Figure 1C* for the L90M mutation in HIV protease. There is a significant performance gain for the Potts model with a true positive rate (TPR) of 0.81 at a false positive rate (FPR) of 0.1, compared to a TPR of 0.46 at an FPR of 0.1 for the Hamming distance model, again showing that the latter model is much less predictive.

### 'Entrenchment' of a primary resistance mutation and its verification

The log likelihood of a mutation occurring at a position $i$ in a given sequence background relative to the wild-type residue (consensus) at that position can be calculated from the change in the Potts statistical energy on acquiring that mutation from the wild-type residue at that site as $\Delta E^i = E^i_{wild-type} - E^i_{mutation}$. If $\Delta E^i > 0$, this implies that the mutation is more favorable in the given

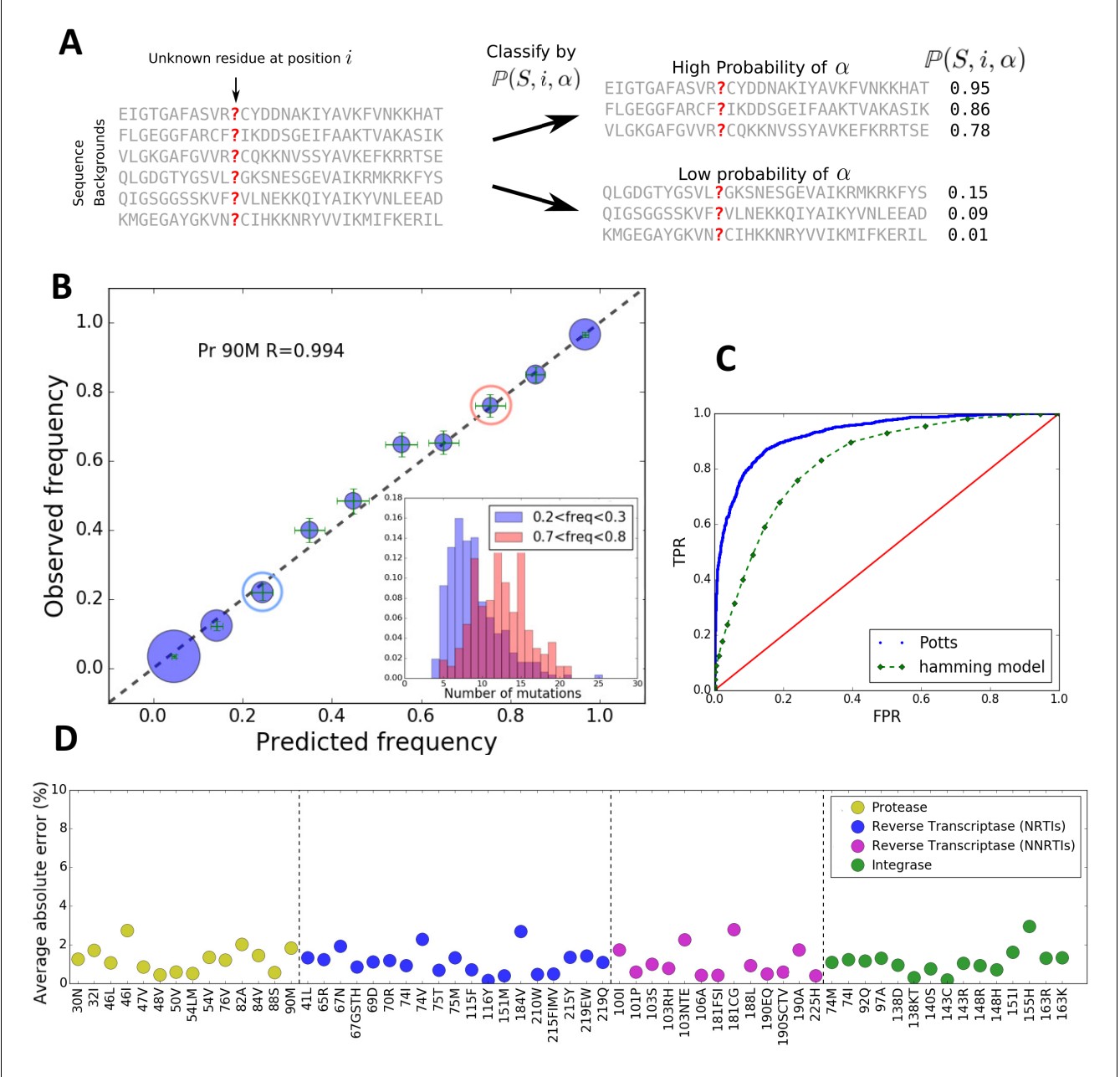

**Figure 1.** The Potts model predicts residue frequencies. (**A**) Schematic showing that the Potts model can be used to classify sequences by how likely a residue $\alpha$ is to appear at a position $i$ in a sequence $S$ using the background-dependent probability, $\mathbb{P}(S, i, \alpha)$. (**B**) The observed frequency of the resistance mutation L90M in HIV-1 drug-experienced proteases matches the Potts model predicted frequencies in sequence clusters binned according to the Potts-predicted frequencies in steps of 0.1 (blue circles) with statistical error (green). Diameters of the circles represent the number of sequences. Inset shows the significant overlap in Hamming distance for sequences with low predicted mutant frequencies between 0.2 and 0.3 (blue) and high, 0.7 and 0.8 (pink) depicting the difficulty of such a classification based on Hamming distance. (**C**) The receiver operator characteristic (ROC) curve comparing the Potts model and Hamming distances as classifiers of mutational probabilities for L90M in HIV protease. (**D**) The average absolute error between the observed mutational frequencies and the Potts-predicted frequencies for the major drug-resistance mutations in HIV-1 in three drug targets. The average absolute error is calculated by binning the sequences in ascending order of their predicted frequencies such that there are roughly equal number of sequences in each bin as shown in *Figure 1—figure supplement 1*, and averaging over the absolute error in each bin.

DOI: https://doi.org/10.7554/eLife.50524.002

The following figure supplement is available for figure 1:

**Figure supplement 1.** Observed vs predicted frequencies for calculating the average absolute error in *Figure 1D*.

DOI: https://doi.org/10.7554/eLife.50524.003

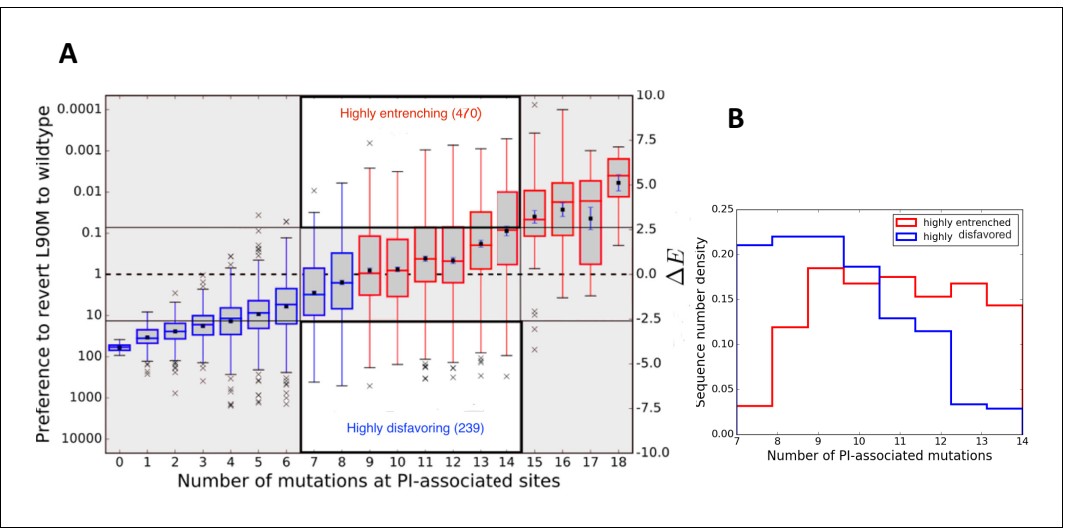

**Figure 2.** Entrenchment and the effect of epistasis on the favorability of a primary resistance mutation. (A) $\Delta E^{90} = E^{90}_{wild-type} - E^{90}_{mutation}$ change for sequences, conditional on the number of PI-associated mutations is shown as boxplots annotating the first, second and third interquartile range. Whiskers extend to 1.5 times the interquartile range with outliers marked as 'x's and the mean values are marked as squares. The left ordinate scale shows the relative probability of reversion ($e^{-\Delta E}$), and the right shows $\Delta E$. Sequences whose energy difference fall above $\Delta E = 0$ (dashed line) are entrenching backgrounds favoring the mutation. Sequence backgrounds where the mutation is favored on average are shown in red, the others in blue. The mutation L90M becomes favorable on average when there are about 9 PI-associated mutations, but there is a wide range of favorability and 'which' PI-associated mutations are present play an important role in determining if the primary mutation is favored/disfavored. The highlighted regions (white with dark border) show there are many sequence backgrounds with between 7 and 14 mutations in which L90M is either 'highly entrenched or favored' (top) or 'highly disfavored' (bottom). (B) Distributions of number of PI-associated mutations in sequences in the 'highly entrenching' and 'highly disfavoring' regions from panel A have a large overlap, again showing that entrenchment is not primarily determined by number of associated mutations. Prediction of the likelihood of M based on the number of PI-associated mutations alone for these sequence backgrounds where M is highly entrenched (red) or highly disfavored (blue) would be especially difficult due to the large overlap between them.

DOI: https://doi.org/10.7554/eLife.50524.004

sequence background and that there is a fitness penalty associated for reversion back to the wild-type residue; while $\Delta E^i < 0$ implies that the wild-type residue is more favorable and the mutation is likely to revert to the wild-type residue. A single resistance mutation in the wild-type background is almost always disfavored relative to the wild-type residue, and is generally disfavored ($\Delta E < 0$) in backgrounds which are close to the wild-type (i.e carry a few accessory mutations). However, as the number of mutations in the background increases, the preference for the wild-type residue at position $i$ decreases on average, until eventually the mutant is preferred at that position. We describe a drug-resistance mutation at position $i$ as being 'entrenched' in a particular sequence S if $\Delta E^i > 0$ at that position. When a drug-resistance mutation is entrenched at $i$ in sequence S, this means that the drug-resistance mutation is more probable than the wild-type residue at $i$ in sequence S.

*Figure 2A* illustrates entrenchment for the drug-resistance mutation L90M in HIV-1 proteases. The preference to revert as a function of the number of mutations at PI-associated sites is shown as box plots. Each box plot shows the median and mean values of the preference to revert for sequences, conditional on the indicated number of mutations at PI-associated sites in the corresponding column, as well as the region containing half of the sequences (shown in gray bounded by blue/red boxes); the whiskers of the box plot indicate the range of $\Delta E$ values for sequences in the tails of the distributions conditional on the number of mutations at PI-associated sites. For drug-experienced viral PR sequence backgrounds that contain eight or fewer PI-associated mutations, (*Figure 2A* left area in blue), the L90M mutation is disfavored on average and likely to revert, with more than a 10 fold preference to revert in backgrounds with no inhibitor-associated mutations. However, the

box plots show that the sequences, conditional on a fixed number of PI-associated mutations, typically span a large range of $\Delta E$s. For example, the mean and median values of $\Delta E$ for L90M in those sequences which contain exactly eight PI-associated mutations is slightly smaller than zero (meaning a probability of reverting slightly greater than one), the range of $\Delta E$ values spans from about $-6$ to $+5.4$ (the preference to revert spans the range $\sim$217x to 0.003x). *Figure 2A* shows that for each box plot corresponding to nine or more PI-associated mutations, the mean and median value of $\Delta E>0$, again for each box plot $\Delta E$ or equivalently, the preference to revert, assumes a large range. It is noteworthy that in the consensus background (one of the sequences containing no PI-associated mutations), the predicted probability of observing the consensus residue L90 is $\sim$40x greater than observing the drug-resistance mutation 90M in the Stanford database of drug-experienced sequences. This prediction for the relative likelihood of L90M in the consensus background is qualitatively consistent with what is actually observed in the drug-experienced Stanford dataset (40 sequences are observed to contain L90 and 2 sequences are observed to contain 90M). *Figure 2* does not however, represent a time ordering for the acquisition of the primary DRM, instead it follows the general favorability and likelihood of the mutation in sequence backgrounds conditional on the number of inhibitor-associated mutations; and the mutation L90M *on average*, is favored over the wild-type residue and is entrenched (unlikely to revert) only in sequence backgrounds with 9 or more PI-associated mutations (seen in *Figure 2A* for sequences in red).

For a quantitative comparison of the Potts model predictions of 'entrenchment' with the observed statistics concerning the relative likelihood of L90M in the Stanford drug-experienced dataset, it is necessary to aggregate the sequence statistics since the dataset size is too small to compare predictions with observations on a sequence by sequence basis. We compare the observed and predicted residue frequencies for different subsets of sequences as shown in *Table 1* for L90M in HIV protease. The agreement between the predictions of the Potts model and the observed sequence statistics is very good. For the set of highly entrenching sequences ($\Delta E^i>1\sigma$ with $\sigma$ being the standard deviation in $\Delta E$) the L90M mutation is predicted to be present in this set with frequency 97.6% whereas it is observed to be present with frequency 97.3% (see *Table 1* second row). In the backgrounds classified by the Potts model as 'highly disfavoring', the mutation is predicted to occur with frequency 3.5% whereas the observed frequency is 1.9%. The L90M mutation is observed to be enriched in the sequences classified by the Potts model as 'highly entrenching' over those classified as 'highly disfavoring' the mutation by a factor of $\sim$1907.

Entrenchment cannot be predicted based simply on the number of mutations, particularly for sequences with an intermediate number of associated mutations in the background. To illustrate this we consider sequences for L90M in HIV-1 PR, with between 7 and 14 inhibitor-associated mutations which correspond to the two highlighted regions (white with dark border) in the center of *Figure 2A* which contain roughly equal numbers of sequences (see *Table 1* third row agreement between predictions and observations is excellent). There is significant overlap of the number of PI-associated mutations in the two distributions as seen in *Figure 2B*, which makes clear why it is difficult to classify them based on the number of associated mutations alone. This also serves to emphasize that

**Table 1.** Confirmation of the Potts model entrenchment predictions for the mutation L90M in HIV-1 protease.
Analytical predictions of likelihoods of the mutation (using the Potts model) are shown along with the corresponding observed frequencies in different subsets of our dataset classified as entrenching ($\Delta E>0$) or disfavoring ($\Delta E<0$) for the mutation. The agreement between the predicted and observed frequencies is remarkable and serves as a confirmation for the Potts model entrenchment predictions.

| Sequences | Classification | Number of sequences | Predicted frequency | Observed frequency |
|---|---|---|---|---|
| All | Entrenched | 1475 | 82.7% | 83.9% |
| | Disfavored | 3283 | 12.7% | 11.9% |
| All with $\Delta E \geq 1\sigma$ of $\Delta E = 0$ | Entrenched | 560 | 97.6% | 97.3% |
| | Disfavored | 1444 | 3.5% | 1.9% |
| With between 7 and 14 mutations and $\Delta E \geq 1\sigma$ of $\Delta E = 0$ | Entrenched | 470 | 97.5% | 97.0% |
| | Disfavored | 239 | 2.8% | 2.1% |

DOI: https://doi.org/10.7554/eLife.50524.005

the probability of observing a drug-resistance mutation depends on the specific pattern of mutations that appears in the background and not just the total number of associated mutations in the background; this is also implied from the variance of the Potts $\Delta E$ (lengths of the box and whiskers in *Figure 2A*) for sequences, for instance, with 13 to 17 PI-associated mutations, where some sequence backgrounds with even such high numbers of associated mutations disfavor the primary DRM while others entrench it.

## Comparative entrenchment of resistance mutations in protease, reverse transcriptase, and integrase

Entrenchment of a resistance mutation depends crucially on the specific background in which it is acquired. In this section, we compare the effect of 'background dependence' of primary DRMs in each of the three major HIV drug target proteins, reverse transcriptase, protease and integrase for the four drug classes: nucleoside analog reverse transcriptase inhibitors (NRTIs), non-nucleoside analog reverse transcriptase inhibitors (NNRTIs), protease inhibitors (PIs) and integrase strand transfer inhibitors (INSTIs) (*Table 2*). Due to the large number of primary resistance mutations, we restrict our analysis to only those occurring at ~1% mutation frequencies or more. A mutation is defined to be 'entrenched in the population' if at least 50% of the sequences which *contain that mutation* have a change in Potts statistical energy score $\Delta E > 0$.

If a resistance mutation is 'entrenched in the population', it means that the majority of sequences containing that mutation are unlikely to revert. These sequences would then support the development of drug resistance in the population as it evolves under drug selection pressure. Most drug-resistance mutations against all four classes of drugs are usually present only at low frequencies (much lower than 50% with the one exception of M184V in RT) in the drug-experienced dataset (*Table 2—source datas 1*, *2*, *3*, *4*); but many of these mutations (with the exception of those conferring resistance to the NNRTIs) are highly 'entrenched in the population' of sequences where they do

**Table 2.** Entrenchment in the population (of sequences carrying the mutation) for four major classes of resistance mutations in HIV-1 subtype B: A DRM is entrenched in the population if at least ~50% or more of the sequences containing the mutation entrench it (i.e $\Delta E = E_{wild} - E_{mut} > 0$).

We catalog entrenchment in the population (of sequences carrying the mutation) for all primary DRMs appearing at ~1% frequency or more, and our study reveals mutations in response to NNRTIs are much less entrenched in the population (of sequences carrying the mutation) than others.

|  | NRTIs | NNRTIs | PIs | INSTIs |
|---|---|---|---|---|
| Years in therapy | 32 | 23 | 24 | 12 |
| Number of primary DRMs | 18 | 15 | 13 | 15 |
| DRMs entrenched in the population (of sequences carrying the mutation) | 11 (61.1%) | 1 ( 6.7%) | 10 (76.9%) | 13 (86.7 %) |
| Number of DRMs conferring high-level resistance | 7 | 11 | 7 | 6 |
| High-level resistance DRMs entrenched in the population (of sequences carrying the mutation) | 3 (42.8%) | 1 (9.1 %) | 5 (71.4%) | 5 (83.3 %) |

Source: Results shown in this table are based on calculations for 'entrenchment in the population' (of sequences carrying the mutation) for each primary DRM occurring at ~1% frequency or more in HIV-1 RT, PR, and IN, respectively as shown in **Table 2—source data 1**, **Table 2—source data 2**, **Table 2—source data 3** and **Table 2—source data 4**. Resistance levels are determined according to Stanford HIVDB (**Rhee et al., 2003**; **Shafer, 2006**) mutation scores for PIs, NRTIs, NNRTIs, and INSTIs.
DOI: https://doi.org/10.7554/eLife.50524.006

The following source data is available for Table 2:
**Source data 1.** Table showing entrenchment in the population (of sequences carrying the mutation) for primary resistance mutations against NRTIs.
DOI: https://doi.org/10.7554/eLife.50524.007
**Source data 2.** Table showing entrenchment in the population (of sequences carrying the mutation) for primary resistance mutations against NNRTIs.
DOI: https://doi.org/10.7554/eLife.50524.008
**Source data 3.** Table showing entrenchment in the population (of sequences carrying the mutation) for primary resistance mutations against PIs.
DOI: https://doi.org/10.7554/eLife.50524.009
**Source data 4.** Table showing entrenchment in the population (of sequences carrying the mutation) for primary resistance mutations against INSTIs.
DOI: https://doi.org/10.7554/eLife.50524.010

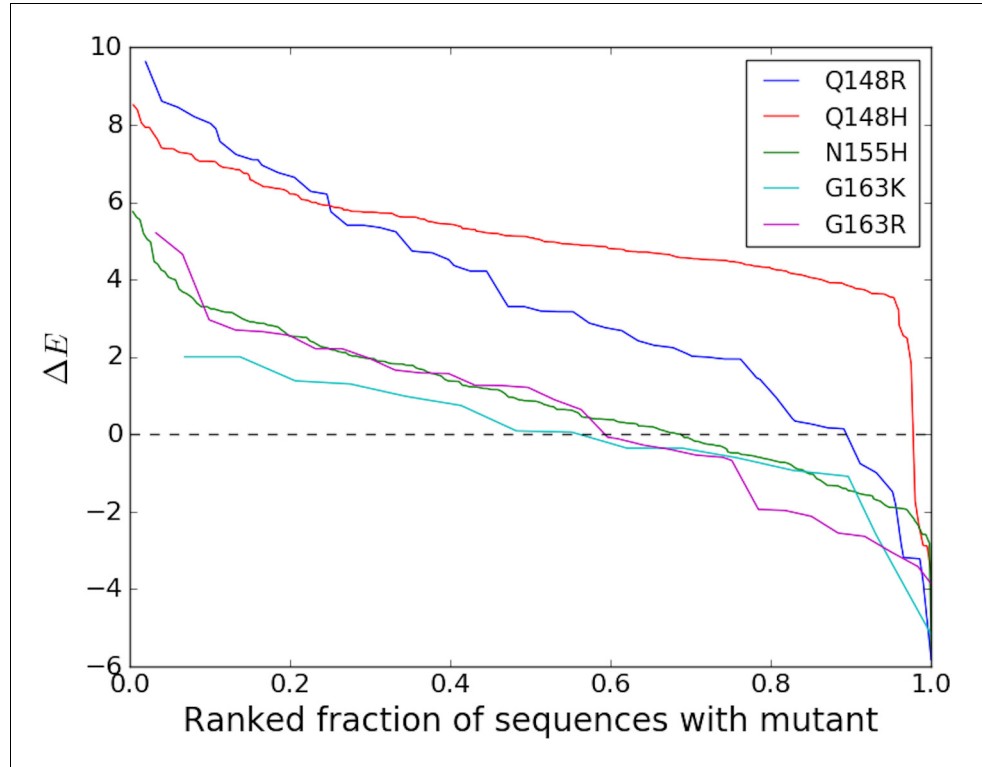

**Figure 3.** Degree of entrenchment of key resistance mutations occurring in the catalytic core domain (CCD) of HIV-1 IN. The change in Potts statistical energy $\Delta E$ for some of the key resistance mutations occurring in the catalytic core domain (CCD) of IN, is plotted as a function of the rank of mutation-carrying sequences, ranked in descending order of their favorability towards the mutant. Plot shows the degree of entrenchment for these mutations. For example, Q148R is highly entrenched in almost all sequences carrying it, whereas, G163K is entrenched ($\Delta E > 0$) in only about half of the sequences carrying G163K.

DOI: https://doi.org/10.7554/eLife.50524.011

occur (**Table 2**), meaning that the mutation is more likely than the wild-type residue in these particular backgrounds, with low probability of reversion. The INSTI-selected resistance mutation Q148R in integrase, for example, is present in only about 5% of the patient population (in our dataset), but the mutation is entrenched in more than 85% of these sequences which have the mutation. These entrenching sequences, even though they are only a small fraction (~4.3%) of the total population, can then support the persistence of drug resistance associated with that mutation as the population evolves under drug selection pressure.

Over the last decade, viral IN has emerged as the major new focus of antiretroviral therapy. IN has been subjected to drug pressure for a considerably shorter period of time than other HIV target proteins, yet IN drug-resistance mutations are strongly 'entrenched in the population' of sequences containing the mutation (**Table 2**) similar to DRMs against PIs, and NRTIs which have been used in therapy for much longer. **Figure 3** illustrates the degree (strength or value of $\Delta E$, and frequency or number of times $\Delta E > 0$) of entrenchment for some key resistance mutations occurring in the catalytic core domain (CCD) of IN. The mutations Q148R and Q148H for example, reduce susceptibilities to the IN strand transfer inhibitors (INSTIs) raltegravir (RAL) and elvitegravir (EVG) by 40–100 fold and 5–10 fold, respectively (**Goethals et al., 2010**; **Fransen et al., 2009**; **Abram et al., 2013**; **Goethals et al., 2008**). Together, they are present in ~20% of the treated population, but they are strongly entrenched in the sequences in which they occur, with the probability of reversion being much smaller than 1%. **Figure 3** shows that the strength of entrenchment can be high for many of these mutations. For example, mutations at site 148 are entrenched in almost all sequences with this mutation, ensuring the persistence of drug resistance associated with these mutations. Comparatively, resistance mutations in RT associated with the non-nucleoside analog reverse transcriptase

inhibitors (NNRTIs) are much less 'entrenched in the population' of sequences containing the mutation (*Table 2*). Only 1 (K103N/T) of the 15 mutations rendering resistance to NNRTIs shows any significant 'entrenchment' by the background.

The striking observation that NNRTI-resistance mutations are less entrenched than those for NRTIs, PIs, and INSTIs is noteworthy (*Table 2*). One clear difference is that NNRTIs are the only allosteric inhibitors among the four target classes being examined here. NRTIs, PIs, and INSTIs exert their action by binding to the catalytic active sites of the target enzymes. One characteristic of active sites is that they contain invariant (or at least highly conserved) residues, and it is possible that nearby mutations interact strongly with those and their potential entrenchment may be influenced by the presence of the conserved residues. Another shared characteristic of the PI, RT, and IN active site is their catalytic residues are all acidic (PR: 2 Asp; RT 3 Asp; IN 1 Glu, 2 Asp). The NNRTI-binding site has considerably more sequence variation, with only Trp229 being invariant. Perhaps the overall variability of sequence in this region makes it more difficult for residues to become entrenched. Another key difference is that the variety of conformational states of the NNRTI-binding region is broader than for the active sites. The pocket can be closed, open, hyperextended open, and open with a shift of the primer grip, with varying degrees of solvent exposure of numerous internal hydrophobic residues (*Sarafianos et al., 2009*).

Entrenchment of a DRM makes it hard to escape drug resistance once the mutation is highly favored in its background. *Table 2* shows the 'entrenchment in the population' of individual DRMs in sequences that carry the DRM. On aggregating over all possible DRMS, the fraction of sequences containing at least one primary DRM is greater than 60% for all four classes of DRMS (see *Table 3* second row). When considering the fraction of sequences carrying at least one entrenched DRM, again the fraction is close to 50% or greater for three of the four DRM classes; the prominent exception is NNRTIs for which only 28.1% carry at least one entrenched primary DRM. This implies that the persistence of drug resistance against NNRTIs as a class (compared to the others) is less likely in the population than the other three classes of HIV drugs.

## Fitness and degree of entrenchment of observed resistance mutations: why some mutations are seen and others are not

The evolutionary trajectory of a resistance mutation depends on the balance between the fitness cost of evolving that mutation in its particular background, and the advantage it provides in evading drug pressure. It has been shown for HIV protease that a majority of the possible amino acid substitutions are observed rarely or not at all in isolates, indicating that the protein function is under strong purifying selection (*Boucher et al., 2019*). Observed resistance mutations arising with lower fitness costs which allow the virus to escape drug pressure with the least effect on its ability to propagate infection are therefore expected to become evolutionarily 'entrenched' by their respective sequence-backgrounds. *Figure 4* shows the distribution of likelihoods as measured by the Potts $\Delta E = E_{wild} - E_{mut}$ scores at drug-resistance sites in sequences where the corresponding drug-resistance mutations are present in HIV integrase. The distribution of likelihoods for observed DRMs in sequences where they are present is seen in 'green', while the 'blue' distribution shows the distribution of likelihoods of mutations to other possible residue types in these sequences (containing the

**Table 3.** Entrenchment of at least one primary DRM for each of the four drug classes in HIV-1: Table shows the percentage of drug-experienced sequences which contain at least one primary DRM, and the percentage of drug-experienced sequences which contain at least one primary DRM such that the DRM is entrenched by its respective background.

This is shown separately for resistance mutations occurring in response to each of the four HIV drug classes. A significantly lower percentage (~28%) of the patient sequences carry at least one entrenched resistance mutation conferring resistance to the NNRTIs when compared to other drugs (~50%–65%).

| | NRTIs | NNRTIs | PI | Insti |
|---|---|---|---|---|
| Number of sequences in MSA | 19194 | 19194 | 4758 | 1220 |
| % of total sequences containing at least one primary DRM | 78.5% | 62.4% | 64.8% | 61.7% |
| % of total sequences with at least one entrenched primary DRM | 64.7% | 28.1% | 50.8% | 47.2 % |

DOI: https://doi.org/10.7554/eLife.50524.012

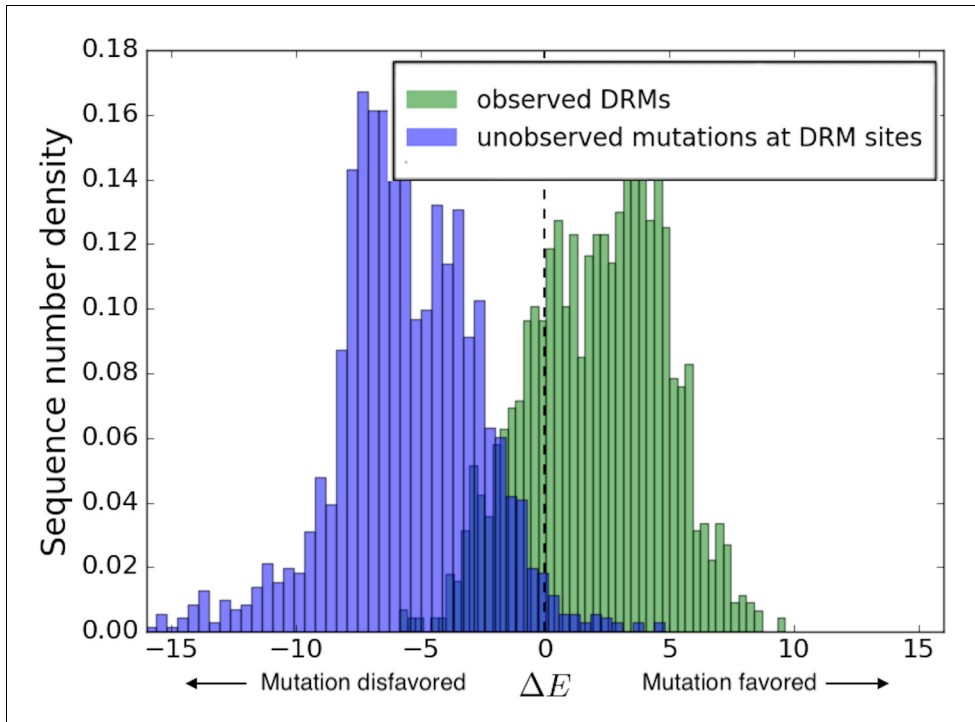

**Figure 4.** Distribution of Potts $\Delta E$ scores for key residues associated with drug resistance in HIV-1 IN. The distribution of the Potts $\Delta E(E_{wild} - E_{mut})$ scores for sequences carrying the particular resistance mutation are shown in 'green' for the most frequently observed INSTI selected resistance mutations in HIV IN, and in 'blue' for all other possible mutations at the same sites. Other possible mutations include rarely observed or unobserved mutations. The histograms show the differential distribution of $\Delta E$ scores for observed vs. unobserved/rare mutations at 15 primary mutation sites associated with evolving drug resistance in HIV-1 IN. The green (observed) and blue (unobserved/rare) distributions are normalized to the total number of primary DRMs in IN in the Stanford HIVDB and the total number of other possible mutations at the same sites, respectively. The mean $\Delta E$ scores for observed vs. unobserved mutations are +2.11 and −5.58, respectively ($p<0.001$). The wide distribution of $\Delta E$ scores also illustrates the role of the background in which the resistance mutation occurs.
DOI: https://doi.org/10.7554/eLife.50524.013

DRMs) at the sites where the DRM is observed. The distribution of the likelihoods of observed IN DRMs (green in *Figure 4*) shows that these drug-resistance mutations are generally more favorable than the wild-type residue as they have a $\Delta E>0$ in their respective backgrounds where the mutations are present. The relative displacement of the green from the blue distribution shows that in sequences which contain the drug-resistance mutation, the observed mutations are predicted on average to be more likely than the consensus residue type and that mutations to other residue types at those sites in the same sequence backgrounds are even less likely than the consensus residue type, on average. Hence, the observed resistance mutations arise in these drug-experienced backgrounds and help enable the virus to evade drug pressure. The more prohibitive fitness costs associated with the appearance of rare/unobserved mutations (blue distribution in *Figure 4*) at the same sites support the suggestion that drug-resistance mutations generally confer resistance at the least fitness cost. Each distribution also has a wide range of $\Delta E$ scores illustrating the wide variation in the favorabilities of different mutations in different sequence backgrounds.

## Molecular clones and the effects of specific backgrounds

In vitro fitness and drug susceptibility assays for viral proteins of HIV are often based on *mutagenesis* experiments, which are performed with specific molecular clones of the virus such as *NL4-3, HXB2, LAI IIIB*, etc (*Hu and Kuritzkes, 2010*; *Abram et al., 2013*). Mutagenesis experiments performed with specific clones, however, can only explore a limited region of the fitness and mutational landscapes available to the viral proteins accessible through the particular clones. These measurements

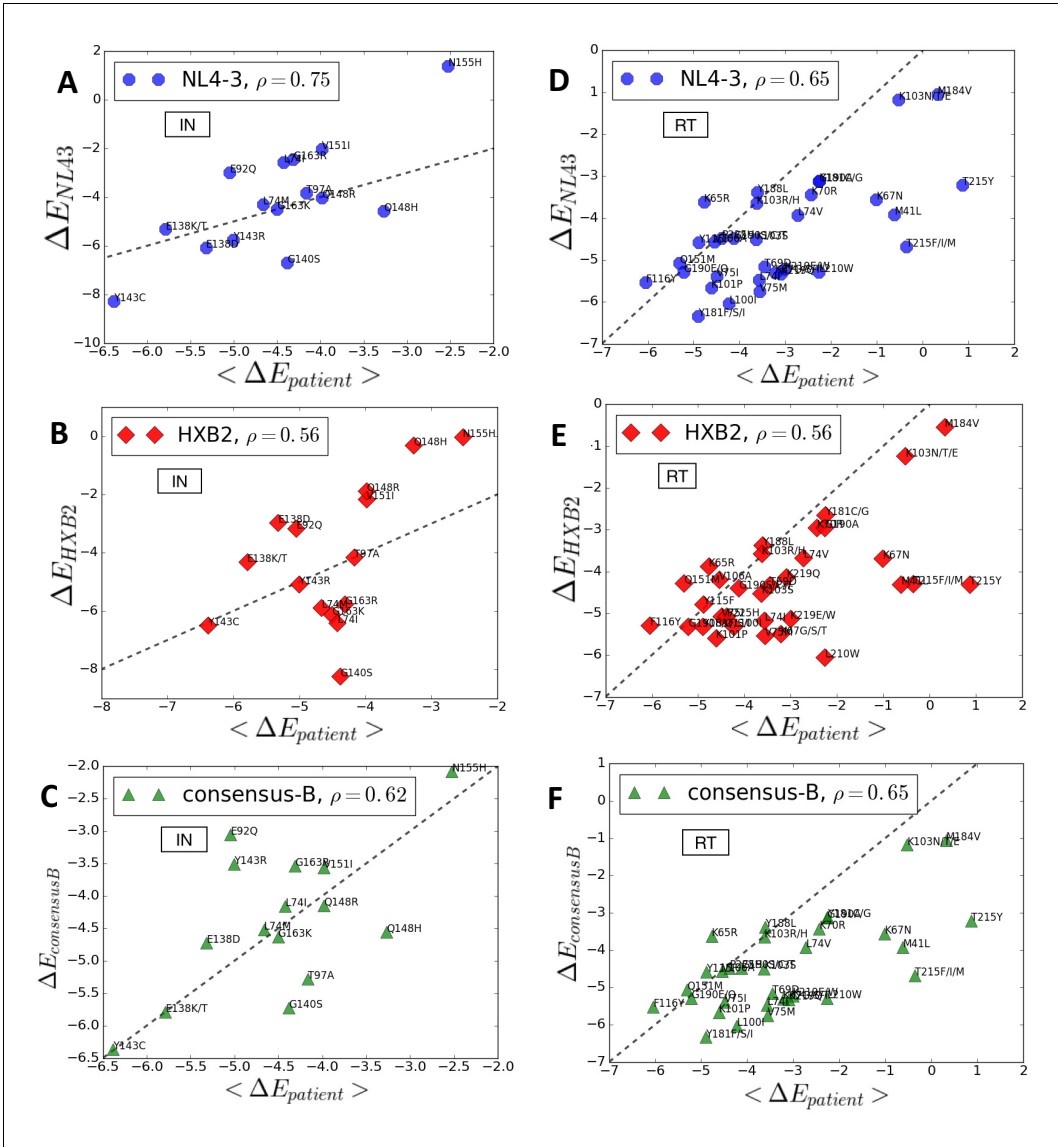

**Figure 5.** Entrenchment and favorability of key resistance mutations in specific backgrounds. The $\Delta E$ change in Potts energy of a sequence is used as the measure of 'entrenchment' and favorability of key resistance mutations in HIV-1 NL4-3, HXB2, and the subtype-B consensus sequences, respectively, shown as a function of the average $\Delta E$ in drug-experienced HIV-1 subtype B patient populations ($<\Delta E_{patient}>$) in the Stanford HIVDB for viral integrase (A,B,C) and reverse transcriptase (D,E,F). In each case, the Pearson correlation coefficients are indicated. The protein sequences for the molecular clones NL4-3 and HXB2 are obtained from GenBank with accession number AF324493.2 and K03455.1, respectively with protein ids for the pol polyprotein as AAK08484.2 and AAB50259.1, respectively. The subtype B consensus sequence is obtained from the Stanford HIVDB. The degree of 'entrenchment' in these subtype B strains is often not representative of the average entrenchment effects in a patient population or even the most representative background from a patient population.
DOI: https://doi.org/10.7554/eLife.50524.014

are not necessarily representative of the fitness effects and the favorabilities of primary resistance mutations in the presence of in vivo selective pressures and entrenchment in viral background(s) of diverse patient populations as contained in the Stanford HIV drug resistance database (HIVDB, https://hivdb.stanford.edu/).

*Figure 5* compares the predicted log likelihoods (Potts $\Delta E$s also associated with the degree of entrenchment) of primary drug-resistance mutations in integrase and reverse transcriptase averaged over the drug-experienced patient sequences in the Stanford HIVDB (denoted as $<\Delta E_{patient}>$) with

the corresponding values in the specific backgrounds of molecular clones NL4-3 (panels A and D), HXB2 (panels B and E) and the subtype B consensus sequence (panels C and F), respectively. The likelihood and degree of entrenchment of a primary resistance mutation depends strongly on the specific background in which it is accrued and as observed in *Figure 5*, these effects studied in the genetic background of a specific molecular clone does not accurately (correlates only moderately) represent the effects averaged over drug-experienced patient populations. Among the three specific sequences analyzed, NL4-3 seems to be somewhat more representative of drug-experienced patient populations than the other two. That the consensus B sequence is not the most representative serves to highlight that epistatic effects can vary significantly from the average to patient populations. In fact, the molecular clones likely represent patient backgrounds where the mutation is disfavored and likely absent, rather than where the mutation is entrenched.

*Figure 6* shows the distribution of the change in Potts energy, $\Delta E$ (log likelihood) values for the DRMS N155H (panel A) and G140S (panel B) in HIV IN in different drug-experienced sequence backgrounds in the Stanford HIVDB along with the $\Delta E$ change in the specific backgrounds of NL4-3, HXB-2 and the subtype B consensus sequences. In terms of the change in Potts energy, which represents the degree of entrenchment of a mutation and its overall favorability in a given background, there is a clear distinction between sequence backgrounds where the mutation is present and mostly favored (or 'entrenched') from those where the mutation is absent. Key resistance mutations are typically disfavored in molecular clones in line with the fitness effects reported in *Hu and Kuritzkes (2010)* and in line with sequence backgrounds that do not carry the mutation (*Figure 6*). An exception to the general rule is observed for the IN mutation N155H in NL4-3 (*Figure 6A*), where the virus, despite not carrying the mutation, is favorable towards it.

## Effective epistasis in the presence of drug selection pressure

HIV evolves rapidly with studies indicating that in the absence of drug pressure, the virus in a single patient explores the majority of all point mutations many times daily (*Coffin, 1995*; *Perelson et al., 1996*). The presence of drug selection pressure can however bias the virus to explore regions of the mutational landscape that allow the virus to evade the drug, which are not accessible otherwise. In this section, we discuss how drug treatment leads to a mutational landscape that reflects both intrinsic epistatic effects and the effects of drug selection pressure; we use the term 'effective epistasis' to refer to the combined effects of both.

As HIV jumps from host to host, it adapts to the applied selection pressure from each host's immune response and drug regimen if the host is currently undergoing retroviral treatment. In the case of immune response selection pressure, (*Shekhar et al., 2013*; *Ferguson et al., 2013*) suggest that due to the diversity of host immune responses among the HIV population, selective effects of immune pressure are averaged so that a Potts model fit to sequences from many different hosts can effectively capture the 'intrinsic' fitness landscape of the virus. In the case of drug selection pressure, the correlated mutation patterns which are contained in MSAs built on drug-experienced datasets reflect a combination of intrinsic epistatic effects and possibly correlations induced by drug selection pressure. To distinguish between intrinsic epistasis and correlated drug selection pressure, we have built a second Potts statistical energy model on an MSA constructed from the Stanford HIV drug-naive dataset and compared the two models.

*Figure 7* shows that the effects of point mutations captured by the changes in Potts statistical energies, $\Delta E = E_{wild-type} - E_{mutation}$ when scored using the two models (the drug-naive model fit to an MSA constructed using HIV sequences in the Stanford database without prior drug exposure and the drug-experienced model fit to an MSA constructed with drug-experienced sequences) are highly correlated. The changes in the 'field' terms are responsible for the shift in $\Delta E$s, whereas the changes in the 'couplings' determine the correlation coefficient. Overall, the high correlation (Pearson correlation coefficient > 0.8) between the probabilities of observing the drug-resistance mutation relative to the wild-type in a given background (measured as a function of $\Delta E$), when the Potts model is parameterized on the drug-naive dataset as compared with the drug-experienced dataset shows that 'intrinsic' epistatic effects have a large influence on the virus evolving under drug selection pressure. A detailed comparison of the two models will be presented elsewhere.

As a consequence of fitting a Potts model on drug-experienced sequences, there may be apparent mutational correlations induced due to the varying selective pressures of different drug regimens. However, the large overlap in resistance mutation sites among different drugs of the same

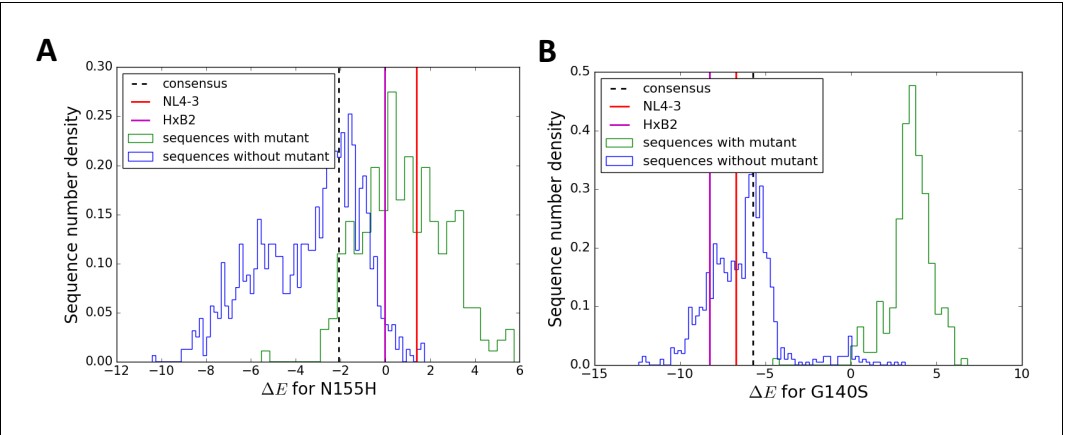

**Figure 6.** The particular sequence background in which a resistance mutation occurs affects the degree of entrenchment, with often a clear distinction between sequences where the mutation is present (green) versus absent (blue) shown here for the mutations N155H (**A**) and G140S (**B**) in integrase. The degree of entrenchment for subtype B consensus, NL4-3 and HXB2 are shown as 'black dashed', 'red' and 'magenta' lines, respectively. The Potts entrenchment score manifests as a clear distinction between backgrounds where a particular mutation is observed from ones where the mutation is absent, with the former more likely to present a distinct fitness advantage towards the mutation.

DOI: https://doi.org/10.7554/eLife.50524.015

class (*Wensing et al., 2017*) as seen in *Figure 8* as well as the large number of possible resistance mutations that can arise in response to any one drug, and the multi-drug resistance mutation patterns induced by many drugs (*Condra et al., 1995*; *Hertogs et al., 2000*; *Delaugerre et al., 2001*), all tend to minimize the effects of such 'spurious' correlations in the drug-experienced dataset. Furthermore, if such 'spurious' correlations picked up from a drug-experienced dataset strongly affected the drug-experienced Potts model, we would not expect to see the 'effective' epistatic effects on $\Delta E$ inferred by the drug-experienced model to be highly correlated with the 'intrinsic' epistatic effects of the drug-naive model as observed in *Figure 7*.

## Discussion

The evolution of viruses like HIV under drug and immune selective pressures induces correlated mutations due to constraints on the structural stability and fitness (ability to assemble, replicate, and propagate infection) of the virus (*Theys et al., 2018*). This is a manifestation of the epistatic interactions in the viral genome. In fact, it has been shown that long-range epistasis can shift a protein's mutational tolerance during HIV evolution (*Haddox et al., 2018*). Epistasis has recently become a major focus in structural biology and genomics, and co-evolutionary information encoded in collections of protein sequences have been used to infer epistatic couplings between them (*Hinkley et al., 2011*; *Ferguson et al., 2013*; *Mann et al., 2014*; *Hopf et al., 2017*; *Figliuzzi et al., 2016*; *Butler et al., 2016*; *Haldane et al., 2018*). In the current study, we have used the correlated mutations encoded in an MSA of drug-experienced HIV-1 sequences to parameterize a Potts Hamiltonian model of sequence statistical energies, and used it to infer the epistatic interactions leading to drug resistance in HIV-1 Subtype B. We first confirmed the Potts model's ability to accurately predict the likelihood of a mutation based on a Potts statistical energy analysis. The predictive power of the model is established by verifying predictions of mutation prevalence based on its background using aggregate sequence statistics from the MSA, illustrating the crucial role of the background in giving rise to a mutation. This also provides the most direct verification of the complex patterns of epistasis which lead to drug resistance in HIV.

Entrenchment of functionally important mutations has been shown to be a general feature in systems exhibiting epistasis (*Pollock et al., 2012*). The entrenchment of primary resistance mutations

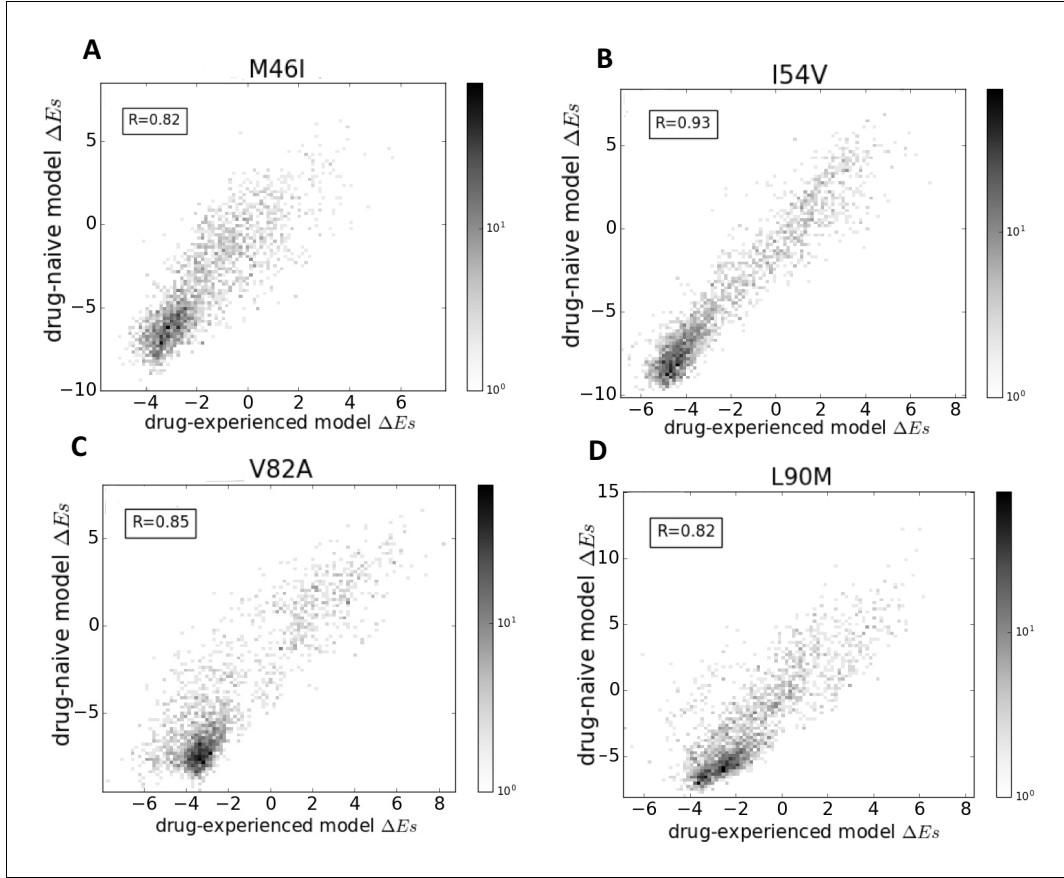

**Figure 7.** Comparison of Potts models parameterized on drug-naive and drug-experienced HIV protein sequences. The comparison of the effects of point mutations is shown in terms of Potts $\Delta E$ scores (which forms the basis of our study) using two different Potts models, parameterized on drug-naive vs drug-experienced sequences for PR drug-resistance mutations M46I (**A**), I54V (**B**), V82A (**C**), and L90M (**D**), all of which appear with at least a frequency of 0.25% in both datasets. Bin shading for the 2D histogram scatter plots shown here scales logarithmically with the number of sequences whose scores fall into each bin. To obtain $\Delta E$ scores, the sequences are scored using a drug-naive model vs. a drug-experienced model. The $\Delta E$ scores are highly correlated with a Pearson correlation coefficient of 0.82 ($p<0.001$), 0.93 ($p<0.001$), 0.85 ($p<0.001$), and 0.82 ($p<0.001$), respectively.
DOI: https://doi.org/10.7554/eLife.50524.016

described in this work suggests that epistasis plays a major role in the evolution of HIV under drug selection pressure. Both primary and accessory drug-resistance mutations exhibit strong epistatic interactions and therefore, entrenchment is a likely mechanism by which drug-resistance mutations accumulate within the population and in the persistence of drug resistance. Building on previous work by *Flynn et al. (2017)* identifying sequence backgrounds which were predicted to strongly entrench or strongly disfavor particular primary resistance mutations in HIV-1 PR, we first confirmed the Potts model entrenchment predictions using aggregate statistics from the MSA, and then extended it to the two other major HIV drug target enzymes, RT and IN.

It has been shown by *Barton et al. (2016b)* and *Chen and Kardar (2019)* that large differences in escape times for HIV patients targeting the same epitope can be explained by differences in the background mutations contained in the sequence of the virus strains that infected these patients. A similar result can also be expected for the virus subjected to antiretroviral therapy. If the background mutations have strong disfavorable couplings with the primary escape mutation, the escape mutation will likely take much longer to establish itself in the population. Due to the nature of entrenchment, we expect this to be reflected in the entrenchment score of a primary mutation in a given background. Resistance mutations, particularly ones associated with high levels of drug resistance, can be expected to occur with a significant degree of entrenchment in most drug-experienced

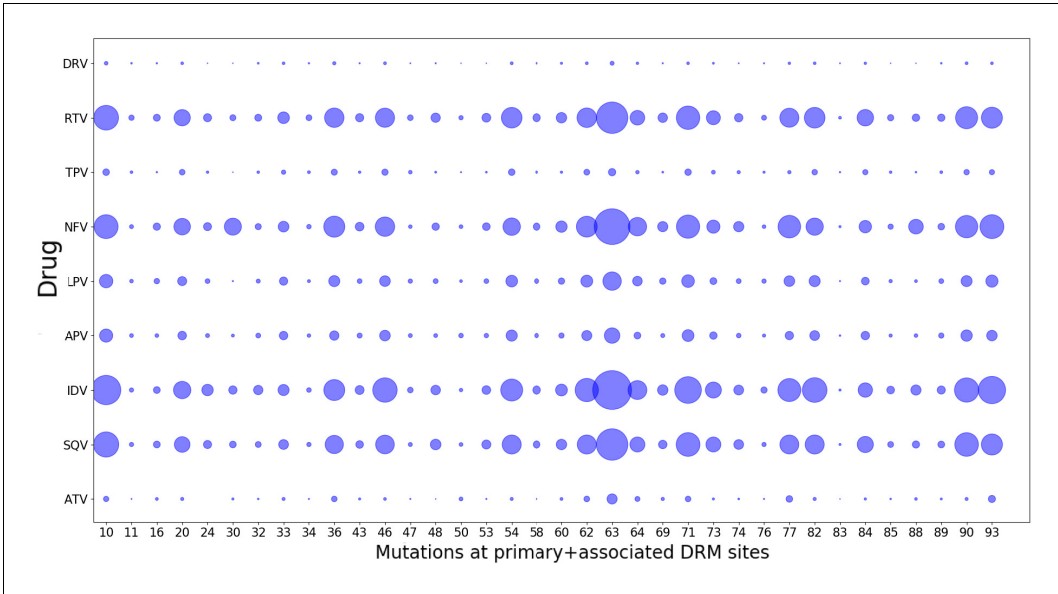

**Figure 8.** Drug-pressure associated mutations are largely common between drugs of the same class. The mutations (both primary and associated) arising in drug-treated HIV proteases in response to inhibitor treatment are shown corresponding to each protease drug. The diameters of the circles represent the number of mutations at that site that occurred in sequences treated with the particular drug. Most mutations that occur in response to one drug are seen to have occurred when treated with another drug of the same class, showing that the 'spurious correlations' (that could be picked up by a Potts model built on a mixture of patient sequences treated with different drugs of the same class, if the mutations occurring in response to one drug are not at all observed in response to another), are minimal.

DOI: https://doi.org/10.7554/eLife.50524.017

backgrounds containing the mutation. Therefore, in the context of patient populations, a study of the entrenchment of primary resistance mutations is very relevant. The most entrenched mutations are the ones which are at some local maxima in the fitness landscape. Accumulating correlated mutations as observed in *Figure 2A*, can unlock pathways to these local fitness maxima (*Gupta and Adami, 2016*). But of crucial importance is the background in which these mutations occur. The local maxima can be 100 times more favorable in particular backgrounds, and these highly resistant sequences pose a significant risk for transmission of drug resistance to new hosts. Entrenchment of resistance mutations can thus play a central role in the persistence of drug-resistant viruses, and many of the most strongly entrenched mutations have been shown to revert slowly in drug-naive patients with transmitted resistance or in drug-experienced patients after withdrawal of ART (*Izopet et al., 2000*; *Yang et al., 2015*; *Gandhi et al., 2003*; *Borman et al., 1996*).

In vitro studies of the acquisition of drug resistance are based on mutagenesis experiments usually performed on molecular clones such as NL4-3, HXB2, IIIB, etc. We show that the epistatic effects on major drug-resistance mutations in molecular clones can be very different from what is observed in patient populations. Thus, fitness studies in particular clones may not be representative of fitness in drug-experienced patient populations or even the consensus background of a patient population.

Our comparison of the statistical energy differences $\Delta E$ obtained from two Potts models of HIV protease sequence co-variation, one constructed from an MSA of drug-naive sequences in the Stanford HIV database, and the other constructed from an MSA of drug-experienced sequences, shows that the effect of the sequence background on the likelihood of observing a drug-resistance mutation compared with a consensus residue is highly correlated between the drug-experienced and drug-naive datasets (*Figure 7*). This supports our conclusion that 'intrinsic' epistatic effects have a strong influence on the virus evolving under drug selection pressure. While beyond the scope of the present work, this observation invites further analysis and opens up new avenues of research designed to interrogate in a quantitative way how the application of drug selection pressure changes the intrinsic mutational landscape. In order to identify HIV sequences which are most responsible for

conferring drug resistance, we need to know the probabilities of observing highly entrenching sequences in the drug-naive population as well as in the drug-experienced population. Multi-canonical reweighting techniques have been developed for this purpose (*Tan et al., 2016*). While the present work does not deal with the time development of drug resistance or identify evolutionary paths, our construction of Potts models from both the drug-naive and drug-experienced populations, sets the stage for kinetic Monte-Carlo studies of pathways by which drug resistance is acquired.

Overall, the analysis presented here provides a framework for further studies of the fitness of HIV proteins evolving under drug selection pressure. The Potts model parameterized on the drug-experienced dataset is a powerful classifier that can identify complex sequence patterns that highly favor (entrench) or disfavor each drug-resistance mutation. Elucidating the epistatic effects of key resistance mutations has the potential to impact future HIV therapies and their entrenchment may be an important factor reinforcing the emergence of drug-resistant strains. The degree of entrenchment of key resistance mutations can provide insights about the fitness of these mutations in the context of a patient HIV reservoir; and their differential degree of entrenchment in different viral sequence backgrounds has the potential to impact future drug design strategies for the next generation of c-ART based on the patient viral reservoir, and suggests that the future of these strategies may very well involve personalized medicine based on an analysis of each patient's individual viral genomic background.

## Materials and methods

In this section, we give a brief introduction to the Potts Hamiltonian model and the motivation behind the model. The Potts model is a probabilistic model of sequence co-variation built on the single and pairwise site amino-acid frequencies of a protein MSA, and aimed to describe the observation probabilities of the various states of the system (sequences in the MSA).

To build the inference model, the goal is to approximate the unknown empirical probability distribution $P(S)$ which best describes HIV-1 sequences $S$ of length $L$, where each residue is encoded in an alphabet $Q$, by a model probability distribution $P^m(S)$ as in *Mora and Bialek (2011)*. We choose the 'least biased' or maximum entropy distribution as the model distribution. Such distributions maximizing the entropy have been previously derived in *Mézard and Mora (2009)*; *Weigt et al. (2009)*; *Morcos et al. (2011)*; *Ferguson et al. (2013)*; *Barton et al. (2016a)* with the constraint that the empirical univariate and bivariate marginal distributions are effectively preserved. We follow a derivation of the maximum entropy model using Lagrange multipliers as in *Mora and Bialek (2011)* and *Ferguson et al. (2013)*. Our maximum entropy model takes the form of an exponential distribution given by:

$$E(S) = \sum_i^L h_{S_i}^i + \sum_{i<j}^{L(L-1)/2} J_{S_iS_j}^{ij} \tag{1}$$

$$P^m(S) \propto exp(-E(S)) \tag{2}$$

where the quantity $E(S)$ is the Potts Hamiltonian determining the statistical energy of a sequence $S$ of length $L$, the model parameters $h_{S_i}^i$ called 'fields' represent the statistical energy of residue $S_i$ at position i in sequence $S$, and $J_{S_iS_j}^{ij}$ are 'couplings' representing the energy contribution of a position pair $i,j$. In this form, the Potts Hamiltonian consists of $LQ$ field parameters $h_{S_i}^i$ and $\binom{L}{2}Q^2$ coupling parameters $J_{S_iS_j}^{ij}$, and for the exponential distribution $P^m \propto exp(-E)$, negative fields and couplings signify favored amino acids. The change in Potts energy due to mutating a residue $\alpha$ at position $i$ in $S$ to $\beta$ is then given by:

$$\Delta E(S_{\alpha\to\beta}^i) = E(S_\alpha^i) - E(S_\beta^i) = h_\alpha^i - h_\beta^i + \sum_{j\neq i}^L J_{\alpha S_j}^{ij} - J_{\beta S_j}^{ij} \tag{3}$$

In this form, a $\Delta E(S_{\alpha\to\beta}^i) > 0$ implies that the residue $\beta$ is more favorable than residue $\alpha$ at position $i$

for the given sequence $S$. If $\alpha$ represents the wild-type residue at $i$ and $\beta$ the mutant, then the mutant is favorable over the wild-type if $\Delta E > 0$ for the change, and *vice versa*.

The methodology followed in this analysis is similar to the one followed in *Flynn et al. (2017)* for HIV-1 PR (for further details on derivation and description of the model parameters see their Materials and methods section as well as the SI). The sample size of the MSA plays a critical role in determining the quality and effectiveness of the model (*Haldane and Levy, 2019*) and we confirm that the models are fit using sufficient data with minimal overfitting.

## Data processing and mutation classification

The method begins with the collection of sequence and patient as well as reference information from the Stanford University HIV Drug resistance database (HIVDB, https://hivdb.stanford.edu) (*Rhee et al., 2003*; *Shafer, 2006*). In this work, we have used the Stanford HIVDB genotype-rx (https: //hivdb.stanford.edu/pages/genotype-rx.html) to obtain protein sequences for HIV target proteins. Alternatively, downloadable sequence datasets are also available on the Stanford HIVDB ( https://hivdb.stanford.edu/pages/geno-rx-datasets.html). The filtering criteria we used are: HIV-1, subtype B and nonCRF, and drug-experienced (# of PI = 1–9 for PR, # of NRTI = 1–9 and # of NNRTI = 1–4 for RT, and # of INST = 1–3 for IN), removal of mixtures, and unambiguous amino acid sequences (amino acids are '-ACDEFGHIKLMNPQRSTVWY'). For RT, sequences with exposure to both NRTIs and NNRTIs were selected. An alternate search for RT sequences exposed to only NRTIs or NNRTIs would return a vastly smaller number of isolates (5398 and 80 isolates, respectively for sequences exposed to only NRTIs or only NNRTIs as compared to 22446 isolates for sequences exposed to both). Sequences with insertions ('#') and deletions ('~') are removed. MSA columns and rows with more than 1% gaps ('.') are removed. This resulted in a final MSA size of $N = 4758$ sequences of length $L = 93$ for PR, $N = 19194$ sequences of length $L = 188$ for RT, and $N = 1220$ sequences of length $L = 263$ for IN. (Sequence datasets now available in the Stanford HIVDB are further updated, last on February, 2019). To retain enough sequence coverage in the MSA, we removed residues: residues 1–38 and residue 227 onwards for RT (see *Appendix 1—figure 1* of Appendix), and 264 onwards for IN. For this reason, some interesting DRMs like F227I/L/V/C, L234I, P236L or N348I (NNRTI affected) for RT are not amenable for our analysis. For drug-naive PR model, sequences are obtained from the Stanford HIVDB using the criterion # of PI = 0, and after filtering of inserts, deletes, rows or columns with too many 'gap' characters, etc results in an MSA of $N = 14969$ sequences of length $L = 93$. The protein sequences for the molecular clones NL4-3 and HXB2 are obtained from GenBank (*Clark et al., 2016*) with accession number AF324493.2 and K03455.1, respectively. The protein ids for the pol polyprotein are AAK08484.2 and AAB50259.1, respectively. The subtype B consensus sequence is obtained from the Los Alamos HIV sequence database (*Foley et al., 2018*) consensus and ancestral sequence alignments (https://www.hiv.lanl.gov/content/sequence/HIV/CONSENSUS/Consensus.html, last updated August 2004), and is also available in Appendix 1 of the Stanford HIVDB release notes page (https://hivdb.stanford.edu/page/release-notes/). Sequences are given weights reciprocal to the number of sequences contributed by each patient such that if multiple sequences are obtained from a single patient, the effective number of sequences obtained from any single patient is still 1. Sequences from different patients are assumed to be independent. It has been shown that phylogenetic trees of drug-naive and drug-experienced HIV-1 patients exhibit star-like phylogenies (*Keele et al., 2008*; *Gupta and Adami, 2016*), and thus, phylogenetic corrections are not required. Potts models of different HIV-1 protein sequences under immune pressure have also previously been parameterized without phylogenetic corrections (*Ferguson et al., 2013*; *Mann et al., 2014*; *Butler et al., 2016*).

In this work, mutations in the HIV genome have been classified into three main classes: primary or signature drug-resistance mutations, accessory or secondary mutations, and polymorphic mutations. The subtype B consensus sequence, which is derived from an alignment of subtype B sequences maintained at the Los Alamos HIV Sequence Database (https://www.hiv.lanl.gov), and is a commonly used reference sequence to which new sequences are compared, is used as our reference sequence and is referred to as the 'consensus wild-type' throughout the text. In accordance with current literature *Wensing et al. (2017)* and the Stanford HIV Drug Resistance Database (*Rhee et al., 2003*; *Shafer, 2006*), any mutation that (i) affects in vitro drug-susceptibility, (ii) occurs commonly in persons experiencing virological failure, and (iii) occurs with fairly low extent of polymorphism (mutations occurring commonly as natural variants) among untreated individuals, is classified as a major or

primary drug-resistance mutation. In contrast, mutations with little or no effect on drug susceptibility are classified as accessory or secondary. Such mutations may reduce drug susceptibility or increase replication fitness only in combination with a primary mutation. Polymorphic mutations are mutations occurring as natural variants typically in antiretroviral drug (ARV) naive patients. Polymorphic mutations may occur in the absence of drug pressure and usually have little effect on ARV susceptibility when they occur without other DRMs. In this regard, polymorphic mutations affecting ARV susceptibility in combination with other DRMs can be classified as accessory.

## Alphabet reduction

It has been shown that a reduced grouping of alphabets based on statistical properties can still capture most of the information in the full 20 letter alphabet while decreasing the number of degrees of freedom and thereby, leading to a more efficient model inference (*Barton et al., 2016a*; *Haldane et al., 2016*). All the possible alphabet reductions from 21 amino acid characters (20+1 gap) to 20 amino acid characters at any site $i$ are enumerated for all pair of positions $ij(j \neq i)$, by summing the bivariate marginals for each of the $\binom{21}{2}$ possible combinations of amino acid characters, and choosing the alphabet grouping which minimizes the root mean square difference (RMSD) in mutual information (MI):

$$MI_{RMSD} = \sqrt{\frac{1}{N}\sum_{ij}\left(MI_{Q=21}^{ij} - MI_{Q=20}^{ij}\right)^2} \qquad (4)$$

between all pairs of positions $ij(j \neq i)$. The process is then iteratively carried out from 20 amino acid characters to 19, and so on until the desired reduction in amino acid characters is reached. Due to residue conservation at many sites in HIV-1, several previous studies have used a binary alphabet to extract meaningful information from HIV sequences (*Wu et al., 2003*; *Ferguson et al., 2013*; *Flynn et al., 2015*). A binary alphabet however, marginalizes the information at a site to only the wild-type and mutant residues, with the loss of some informative distinctions between residues particularly at sites acquiring multiple mutations. To strike a balance between loss of information due alphabet reduction and reduction of the number of degrees of freedom for effective model inference, we use a reduced alphabet of 4 letters in line with *Flynn et al. (2017)*. Our four letter alphabet reduction gives a *Pearson's $R^2$* coefficient of 0.995, 0.984, and 0.980 for protease, reverse transcriptase, and integrase, respectively between the *MI* of bivariate marginal distributions with the full 21 letter alphabet and the reduced four letter alphabet, representing a minimal loss of information due to the reduction.

Using the reduced alphabet, the original MSA is then re-encoded and the bivariate marginals are recalculated. Small pseudocounts are then added to the bivariate marginals, as described by *Haldane et al. (2016)* to make up for sampling bias or limit divergence in the following inference procedure.

## Model inference

The goal of the model inference is to find a suitable set of Potts parameters $\{h, J\}$ that fully determines the Potts Hamiltonian $E(S)$ and the total probability distribution $P^m(S)$ given in *Equation 1* and *Equation 2*, respectively. This is done by obtaining the set of fields and couplings $\{h, J\}$, which yield bivariate marginal estimates that best reproduce the empirical bivariate marginals.

A number of techniques to this effect have been developed previously (*Mézard and Mora, 2009*; *Weigt et al., 2009*; *Balakrishnan et al., 2011*; *Cocco and Monasson, 2011*; *Morcos et al., 2011*; *Haq et al., 2012*; *Jones et al., 2012*; *Ekeberg et al., 2013*; *Ferguson et al., 2013*; *Barton et al., 2016a*). We follow the methodology given in *Ferguson et al. (2013)*. Given a set of fields and couplings, the bivariate marginals are estimated by generating sequences through a Markov Chain Monte Carlo (MCMC) sampling with the Metropolis criterion for a generated sequence proportional to the exponentiated Potts Hamiltonian. A multidimensional Newton search algorithm is then used to find the optimal set of Potts parameters $\{h, J\}$. The descent step in the Newton search is determined after comparing the bivariate marginal estimates generated from the MCMC sample with the empirical bivariate marginal distribution. Although approximations are made in the computation of the Newton steps, the advantage of this method is that it avoids making explicit approximations to

the model probability distribution. The method is limited by the sampling error of the input empirical marginal distributions and can also be computationally quite intensive. Our GPU implementation of the MCMC method makes it computationally tractable without resorting to more approximate inverse inference methods. The MCMC algorithm implemented on GPUs has been used to infer Potts models of sequence covariation which are sufficiently accurate to infer higher order marginals as shown by *Flynn et al. (2017)*; *Haldane et al. (2018)*. For a full description of the inference technique, we refer the reader to the supplemental information of *Haldane et al. (2016)*.

The scheme for choosing the Newton update step is important. *Ferguson et al. (2013)* developed a quasi-Newton parameter update approach which determines the updates to $J^{ij}$ and $h^i$ by inverting the system's Jacobian. In order to simplify and speed up the computation, we take advantage of the gauge invariance of the Potts Hamiltonian. We use a fieldless gauge in which $h^i = 0$ for all $i$, and we compute the expected change in the model bivariate marginals $\Delta f_m^{ij}$ (hereafter dropping the m subscript) due to a change in $J^{ij}$ to the first order by:

$$\Delta f_{S_i S_j}^{ij} = \sum_{kl, S_k S_l} \frac{\partial f_{S_i S_j}^{ij}}{\partial J_{S_k S_l}^{kl}} \Delta J_{S_k S_l}^{kl} + \sum_{k, S_k} \frac{\partial f_{S_i S_j}^{ij}}{\partial h_{S_k}^k} \Delta h_{S_k}^k \tag{5}$$

Computing the Jacobian $\frac{\partial f_{S_i S_j}^{ij}}{\partial J_{S_k S_l}^{kl}}$ and inverting the linear system in *Equation 5* to solve for the changes in $\Delta J^{ij}$ and $\Delta h^i$ given the $\Delta f^{ij}$, is the challenging part of the computation. We choose the $\Delta f^{ij}$ as:

$$\Delta f^{ij} = \gamma (f_{empirical}^{ij} - f^{ij}) \tag{6}$$

with a small enough damping parameter $\gamma$ such that the linear (and other) approximations are valid.

The computational cost of fitting $\binom{L}{2} \times (4-1)^2 + 93 \times (4-1) = 372,816$ model parameters for the smallest protein in our analysis, PR, on 2 NVIDIA K80 or 4 NVIDIA TitanX GPUs is $\approx 20\,h$. The methodology followed in this analysis is almost the same as done in *Flynn et al. (2017)* (see Materials and methods) for HIV-1 PR. For a more detailed description of data preprocessing, model inference, and comparison with other methods, we refer the reader to the SI and text of *Flynn et al. (2017)*; *Haldane et al. (2016)*; *Haldane et al. (2018)*; *Haldane and Levy (2019)*.

A repository containing the inference methodology code is available at https://github.com/ComputationalBiophysicsCollaborative/IvoGPU and the final MSAs are available at https://github.com/ComputationalBiophysicsCollaborative/elife_data (copy archived at https://github.com/elifesciences-publications/elife_data). Appendix 1 contains a figure showing the sequence coverage for RT and tables of most entrenched and least entrenched sequences with same Hamming distances, for some important DRMs in PR, RT and IN.

## Statistical robustness of HIV potts models

Finite sampling error and overfitting can play an important role in all inference problems, and inverse Ising inference is no exception. Overfitting of the Potts model cannot be simply estimated by comparing the number of model parameters to the number of sequences being fit. Our Potts model is not directly fit to the sequences but to the bivariate marginals (pairwise residue frequencies) of the MSA, and there are equal numbers of model parameters as there are bivariate marginals. The inference problem is neither over nor under constrained. The major source of error that affects our model inference is the fact that the inference relies on pair statistics (bivariate marginals) found in a finite collection of sequences. Overfitting comes from this finite-sampling error in the bivariate marginals estimated from a finite-sized multiple sequence alignment (MSA) which then serves as the input for model inference. It has been shown by *Haldane and Levy (2019)* that it is possible to fit accurate Potts models to MSAs when the number of sequences in the MSA is much smaller than the number of parameters of the Potts model.

The degree of Potts model overfitting can be quantified using the 'signal to noise ratio', or SNR, which is a function of the sequence length $L$, alphabet size $q$, number of sequences in the MSA $N$, and a measure of the degree of conservation of the MSA, $\chi^2$. Using an SNR analysis of the kind described in *Haldane and Levy (2019)*, we conclude that the Potts models for HIV-1 PR and RT are

clearly not overfit (SNR values are 21.6 for PR, and 43.7 for RT). Since the number of sequences in the IN MSA is considerably smaller than the others, the IN model can be more affected by overfitting and may be less reliable than others (SNR is 0.14). However, the different types of predictions based on Potts models are differently affected by finite sampling error; predictions of the effect of point mutations to a sequence, $\Delta E$, which forms the basis of this study are the most robust and least affected by finite sampling. Using the in silico tests suggested in *Haldane and Levy (2019)*, we find that the effects of point mutations are accurately captured even in the IN model, which is the most susceptible to finite sampling errors among our three models (for a more detailed analysis of the effects of finite sampling on the predictions of the Potts model, we refer the reader to *Haldane and Levy, 2019*). Thus, we conclude that the MSA sample sizes for PR, RT, and IN used in this study are sufficiently large to construct Potts models for these HIV proteins that adequately reflect the effects of the sequence background on point mutations which are the central focus of this work.

## Acknowledgements

We thank the very supportive collaborative environment provided by the HIV Interaction and Viral Evolution (HIVE) Center at the Scripps Research Institute (http://hivescripps.edu, last accessed January, 2019). We thank Alan Engelman for his comments and suggestions for revisions of a draft of this manuscript.

## Additional information

### Funding

| Funder | Grant reference number | Author |
|---|---|---|
| National Institutes of Health | U54-GM103368 | Ronald M Levy<br>Eddy Arnold |
| National Institutes of Health | R01-GM030580 | Ronald M Levy |
| National Institutes of Health | S10OD020095 | Ronald M Levy |
| National Institutes of Health | 1R35GM132090 | Ronald M Levy |

The funders had no role in study design, data collection and interpretation, or the decision to submit the work for publication.

### Author contributions

Avik Biswas, Allan Haldane, Conceptualization, Data curation, Formal analysis, Writing—original draft, Writing—review and editing; Eddy Arnold, Formal analysis, Writing—original draft; Ronald M Levy, Conceptualization, Formal analysis, Writing—original draft, Writing—review and editing

### Author ORCIDs

Avik Biswas (iD) https://orcid.org/0000-0002-3519-3944
Allan Haldane (iD) https://orcid.org/0000-0002-8343-1994
Ronald M Levy (iD) https://orcid.org/0000-0001-8696-5177

### Decision letter and Author response

Decision letter https://doi.org/10.7554/eLife.50524.029
Author response https://doi.org/10.7554/eLife.50524.030

## Additional files

### Supplementary files

• Transparent reporting form
DOI: https://doi.org/10.7554/eLife.50524.018

## Data availability

Sequence data analyzed in this study is obtained from the Stanford University HIV drug resistance database (https://hivdb.stanford.edu/), Los Alamos HIV sequence database (https://www.hiv.lanl.gov/content/sequence/HIV/mainpage.html). Source data tables are provided for Table 2.

The following previously published datasets were used:

| Author(s) | Year | Dataset title | Dataset URL | Database and Identifier |
|---|---|---|---|---|
| Rhee S-Y, Gonzales MJ, Kantor R, Betts BJ, Ravela J, Shafer RW | 2003 | Stanford University HIV drug resistance database: Genotype-Treatment Correlations | https://hivdb.stanford.edu/pages/geno-rx-datasets.html | Stanford HIV drug resistance database, GENOTYPE-RX |
| Foley B, Leitner T, Apetrei C, Hahn B, Mizrachi I, Mullins J, Rambaut A, Wolinsky S, Korber B | 2004 | Consensus and Ancestral Sequence Alignments, Select 'Alignment type:Consensus/Ancestral', 'organism: HIV-1/SIVcpz', 'Pre-defined region of the genome: POL', Subtype:All', 'DNA/PRotein: Protein' | https://hiv.lanl.gov/content/sequence/NEWALIGN/align.html | Los Alamos HIV sequence database, Consensus and Ancestral Sequence Alignments |

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

# Appendix 1

DOI: https://doi.org/10.7554/eLife.50524.019

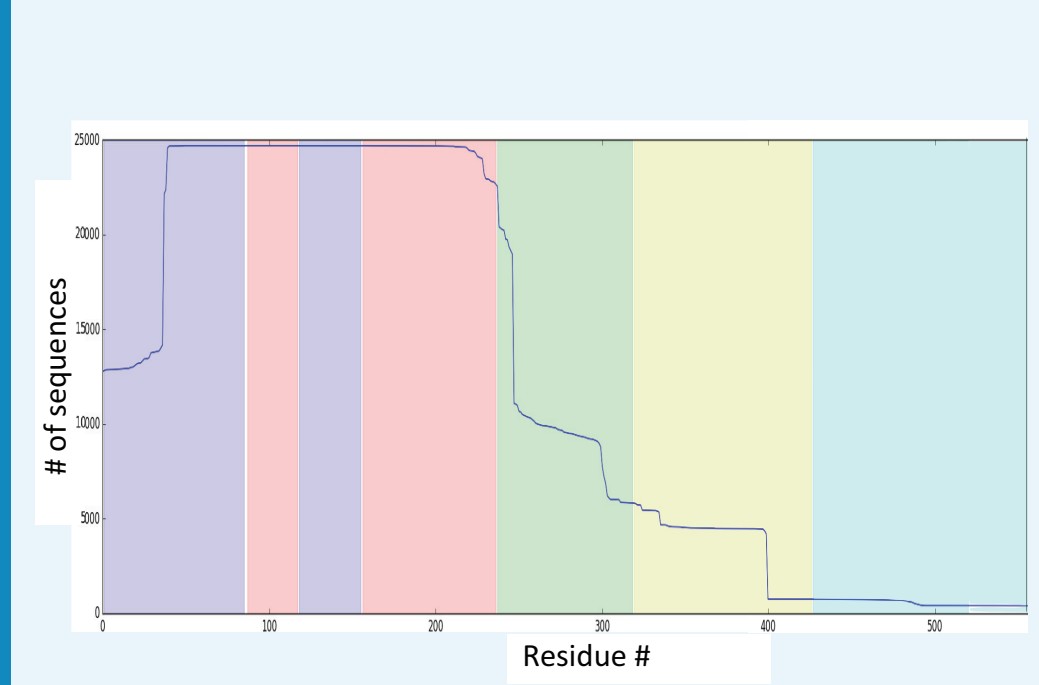

Residue positions are colored by respective domains and subdomains of the p66 subunit:

**Polymerase:**
- Fingers (1-85, 118-155)
- Palm (86-117, 156-236)
- Thumb (237-318)
- Connection (319-426)

**RNase H:**

**Appendix 1—figure 1.** Sequence coverage for RT. Figure shows the sequence coverage (# of sequences vs the # of residues) for RT drug-experienced (both NRTI and NNRTI) sequences derived from the Stanford HIVDB (22,444 isolates from 20422 patients). For RT, sequences with exposure to both NRTIs and NNRTIs were selected as an alternate search for RT sequences exposed to only NRTIs or NNRTIs would return a vastly smaller number of isolates (5398 and 80, respectively). Sequences with insertions ('#') and deletions (' ~') are removed. MSA columns and rows with more than 1% gaps ('.') are removed. This resulted in a final MSA size of $N = 19194$ sequences from 17130 persons each with length $L = 188$ for RT. To retain enough sequence coverage in the MSA, we removed residues: residues 1–38, and residue 227 onwards for RT. For this reason, some interesting DRMs like F227I/L/V/C, L234I, P236L or N348I (NNRTI affected) for RT are not amenable for our analysis.

DOI: https://doi.org/10.7554/eLife.50524.020

**Appendix 1—table 1.** Most entrenched (ME) and least entrenched (LE) or most disfavoring sequence pairs with the same Hamming distances (HD) for primary resistance mutations against the INSTIs

| | | E92Q | | L74M | | N155H | | Q148H | | G140S | |
|---|---|---|---|---|---|---|---|---|---|---|---|
| Position | Consensus | ME | LE | ME | LE | ME | LE | ME | LE | ME | LE |
| 6 | D | – | – | – | E | – | – | – | – | – | – |

*Appendix 1—table 1 continued on next page*

Appendix 1—table 1 continued

| Position | Consensus | E92Q | | L74M | | N155H | | Q148H | | G140S | |
|---|---|---|---|---|---|---|---|---|---|---|---|
| | | ME | LE | ME | LE | ME | LE | ME | LE | ME | LE |
| 11 | E | – | – | – | – | – | – | – | D | – | D |
| 17 | S | – | – | N | N | N | N | – | T | – | – |
| 20 | R | – | – | – | – | – | – | – | – | K | – |
| 22 | M | – | – | I | – | – | – | – | – | – | – |
| 23 | A | – | – | V | – | – | – | – | – | – | – |
| 28 | L | – | – | I | – | – | – | – | – | – | – |
| 31 | V | – | – | – | – | I | – | I | – | I | – |
| 32 | V | – | – | – | – | – | – | – | – | – | I |
| 37 | V | – | – | – | – | – | – | – | – | – | I |
| 39 | S | C | C | – | – | – | – | – | – | – | – |
| 45 | L | Q | – | V | – | – | – | – | – | – | – |
| 50 | M | – | – | M | I | – | – | I | – | L | – |
| 63 | L | I | – | – | – | – | – | – | – | – | – |
| 72 | I | L | – | – | V | – | V | – | – | – | – |
| 74 | L | M | – | M | M | – | – | – | – | – | – |
| 92 | E | Q | Q | – | – | – | – | – | – | – | – |
| 97 | T | – | – | A | A | – | – | – | – | – | – |
| 101 | L | – | I | I | – | I | – | – | I | I | I |
| 119 | S | – | – | T | R | – | – | – | P | – | – |
| 124 | T | – | N | – | – | – | A | – | N | – | – |
| 135 | I | – | V | – | – | – | – | – | V | – | – |
| 136 | K | – | – | – | – | – | – | – | – | Q | – |
| 138 | E | – | – | D | – | – | – | K | – | – | – |
| 140 | G | – | – | – | – | – | S | S | – | S | S |
| 143 | Y | – | – | R | – | – | – | – | – | – | – |
| 148 | Q | – | – | – | – | – | H | H | H | H | R |
| 151 | V | – | – | – | I | I | – | – | – | – | – |
| 154 | M | – | – | – | – | – | I | – | – | – | – |
| 155 | N | – | H | – | H | H | H | – | – | – | – |
| 156 | K | – | – | – | N | – | – | – | – | – | – |
| 170 | E | – | – | – | – | – | A | – | – | – | – |
| 181 | F | – | L | – | – | – | – | – | L | – | – |
| 188 | K | – | – | – | – | – | – | R | – | – | – |
| 196 | A | – | – | – | – | P | – | – | – | – | – |
| 201 | V | – | I | – | – | I | I | I | I | – | I |
| 206 | T | – | – | – | – | – | – | S | – | – | – |
| 208 | I | – | – | – | M | – | – | – | – | L | – |
| 212 | E | – | – | – | A | – | – | – | – | – | – |
| 215 | K | – | – | – | – | – | – | S | – | – | – |
| 216 | Q | N | – | – | – | – | – | – | – | – | – |
| 220 | I | – | – | – | – | – | – | – | L | – | – |

Appendix 1—table 1 continued

| Position | Consensus | E92Q | | L74M | | N155H | | Q148H | | G140S | |
|---|---|---|---|---|---|---|---|---|---|---|---|
| | | ME | LE | ME | LE | ME | LE | ME | LE | ME | LE |
| 230 | S | – | – | – | – | – | – | G | – | – | – |
| 232 | D | – | – | – | – | N | – | – | – | – | E |
| 234 | L | I | – | – | – | – | – | – | – | – | – |
| 256 | D | – | – | E | – | E | – | – | – | E | E |
| HD | | 8 | 8 | 12 | 12 | 9 | 9 | 10 | 10 | 9 | 9 |

Residues same as consensus are shown as '–', mutations are shown as the one letter abbreviated alphabet encoding the mutant residue (in bold are primary drug-resistance mutations appearing at more than 1% frequency).

DOI: https://doi.org/10.7554/eLife.50524.021

**Appendix 1—table 2.** Most entrenched (ME) and least entrenched (LE) or most disfavoring sequence pairs with the same Hamming distances (HD) for primary resistance mutations against NNRTIs

| Position | Consensus | P225H | | Y188L | | L100I | | K103N/E | | Y181C/G ■ | | E138K/R | |
|---|---|---|---|---|---|---|---|---|---|---|---|---|---|
| | | ME | LE | ME | LE | ME | LE | ME | LE | ME | LE | ME | LE |
| 39 | T | – | – | — | – | – | S | – | – | – | – | – | – |
| 41 | M | – | – | L | L | L | L | – | – | – | – | – | L |
| 43 | K | – | – | – | – | – | R | – | – | – | – | – | – |
| 44 | E | – | – | – | D | – | – | – | – | – | – | – | – |
| 48 | S | – | – | – | – | – | – | T | – | – | – | – | – |
| 49 | K | – | – | – | – | – | – | – | R | – | – | – | – |
| 50 | I | – | – | – | – | – | – | – | – | – | – | – | – |
| 58 | T | – | – | N | – | – | – | – | – | – | – | – | – |
| 60 | V | – | – | – | – | – | – | – | – | – | I | – | – |
| 62 | A | – | – | V | – | – | – | – | – | V | – | – | – |
| 65 | K | – | – | – | – | – | – | – | – | R | – | – | – |
| 67 | D | – | – | – | G | N | – | – | – | – | – | – | – |
| 68 | S | – | – | – | G | – | – | – | – | G | – | – | – |
| 69 | T | – | – | N | – | N | – | – | – | I | N | – | – |
| 70 | K | – | – | – | T | – | – | – | – | – | R | – | – |
| 74 | L | – | V | – | V | V | – | V | – | – | – | – | V |
| 75 | V | – | – | I | – | T | – | – | – | A | – | – | – |
| 77 | F | – | – | L | – | – | – | – | – | – | – | – | – |
| 82 | K | – | – | – | – | – | – | – | – | – | – | R | – |
| 90 | V | – | – | – | – | – | – | I | – | I | – | – | – |
| 98 | A | S | G | – | – | – | S | – | S | – | – | – | – |
| 100 | L | – | – | – | – | I | I | I | – | – | – | – | – |
| 101 | K | Q | – | – | – | – | – | – | – | E | E | E | – |
| 103 | K | N | – | – | N | N | – | N | N | – | N | R | – |
| 104 | K | – | – | – | – | – | – | – | – | N | – | – | – |
| 106 | V | – | – | I | – | – | – | – | – | A | – | – | – |
| 108 | V | – | – | – | – | – | – | – | – | I | – | – | – |

*Appendix 1—table 2 continued*

| | Consensus | P225H | | Y188L | | L100I | | K103N/E | | Y181C/G | | E138K/R | |
|---|---|---|---|---|---|---|---|---|---|---|---|---|---|
| | | ME | LE | ME | LE | ME | LE | ME | LE | ME | LE | ME | LE |
| 116 | F | – | – | W | – | – | – | – | – | – | – | – | – |
| 118 | V | – | – | – | I | – | – | – | – | – | – | – | – |
| 122 | K | – | – | – | E | E | – | E | – | – | – | E | E |
| 123 | D | – | – | – | – | – | E | – | – | – | – | – | – |
| 126 | K | – | – | – | – | – | – | – | – | – | – | R | – |
| 135 | I | L | – | – | K | M | V | – | – | – | T | – | – |
| 138 | E | – | – | – | – | – | – | – | K | – | – | K | K |
| 139 | T | – | – | – | – | – | – | – | – | – | – | E | – |
| 142 | I | – | – | – | – | – | Q | – | – | – | V | T | – |
| 151 | Q | – | – | M | – | – | – | – | – | – | – | – | – |
| 158 | A | – | S | – | – | – | – | – | – | – | – | – | S |
| 162 | S | – | – | – | – | – | C | – | – | – | – | – | A |
| 165 | T | – | I | – | – | – | K | – | – | – | – | – | – |
| 166 | K | – | – | – | – | – | – | – | Q | – | – | – | – |
| 169 | E | – | – | – | A | – | – | – | – | – | – | – | – |
| 172 | R | – | – | K | – | – | – | – | – | – | – | – | – |
| 173 | K | – | – | N | – | – | – | – | – | – | – | – | – |
| 176 | P | Q | – | – | – | – | – | – | – | – | – | – | – |
| 177 | D | – | – | – | – | – | – | N | – | – | – | – | – |
| 178 | I | – | – | – | – | – | – | M | – | – | – | M | – |
| 179 | V | – | D | – | I | – | D | – | – | – | – | – | – |
| 180 | I | – | – | – | – | – | – | – | – | – | M | – | – |
| 181 | Y | – | C | – | C | – | – | – | – | C | C | – | – |
| 184 | M | V | – | – | – | V | I | I | – | – | V | – | – |
| 188 | Y | – | – | L | L | – | – | – | – | – | L | – | – |
| 189 | V | – | – | I | – | – | – | – | – | – | – | – | – |
| 190 | G | – | A | – | S | – | – | – | A | A | – | – | – |
| 196 | G | E | – | – | – | – | – | – | – | – | – | – | – |
| 197 | Q | – | – | – | – | – | – | – | K | – | – | – | – |
| 200 | T | A | V | A | – | A | A | A | A | – | A | – | A |
| 202 | I | – | – | V | – | V | – | – | – | – | – | – | – |
| 203 | E | — | – | – | – | D | – | – | – | – | – | – | – |
| 207 | Q | – | – | – | – | – | A | – | – | – | – | – | – |
| 210 | L | – | – | – | W | W | – | – | – | – | – | – | – |
| 211 | R | – | – | – | K | – | K | – | – | K | K | – | K |
| 214 | F | – | – | L | – | – | – | – | – | – | L | – | – |
| 215 | T | – | – | Y | Y | Y | D | – | – | – | – | – | Y |
| 219 | K | – | – | – | – | E | – | – | – | Q | Q | – | – |
| 221 | H | – | – | – | – | – | – | – | – | Y | – | – | – |
| 223 | K | – | – | T | – | – | – | – | – | – | – | – | – |
| 225 | P | H | H | – | – | – | – | H | – | – | – | – | – |
| HD | | 9 | 9 | 18 | 18 | 16 | 16 | 11 | 11 | 14 | 14 | 9 | 9 |

*Appendix 1—table 2 continued*

| Consensus | P225H | | Y188L | | L100I | | K103N/E | | Y181C/G | | ■ | E138K/R | |
|---|---|---|---|---|---|---|---|---|---|---|---|---|---|
| | ME | LE | ME | LE | ME | LE | ME | LE | ME | LE | | ME | LE |

Residues same as consensus are shown as '–', mutations are shown as the one letter abbreviated alphabet encoding the mutant residue (in bold are primary drug-resistance mutations appearing at more than 1% frequency).

DOI: https://doi.org/10.7554/eLife.50524.022

**Appendix 1—table 3.** Most entrenched (ME) and least entrenched (LE) or most disfavoring sequence pairs with the same Hamming distances (HD) for primary resistance mutations against PIs

| Position | Consensus | I50V | | I84V | | L90M | | V82A | |
|---|---|---|---|---|---|---|---|---|---|
| | | ME | LE | ME | LE | ME | LE | ME | LE |
| 10 | L | I | – | F | – | I | – | I | I |
| 13 | I | – | V | – | – | V | – | – | V |
| 15 | I | – | V | – | V | – | V | – | – |
| 18 | Q | – | – | – | – | H | – | – | H |
| 19 | L | – | I | – | – | – | – | – | – |
| 20 | K | – | – | – | – | I | R | – | – |
| 24 | L | – | – | – | – | – | – | I | – |
| 30 | D | – | – | – | N | – | – | – | – |
| 32 | V | – | – | – | – | – | I | – | – |
| 33 | L | – | F | – | F | – | – | F | – |
| 34 | E | Q | – | – | – | – | – | – | – |
| 35 | E | – | D | – | D | – | D | – | – |
| 36 | M | – | – | – | L | – | I | – | – |
| 37 | N | – | S | – | E | – | D | – | – |
| 41 | R | – | K | – | – | – | K | – | – |
| 43 | K | – | – | – | T | – | – | – | – |
| 46 | M | I | – | I | – | – | L | L | I |
| 47 | I | – | – | – | – | – | A | – | – |
| 48 | G | V | – | – | – | – | – | – | – |
| 50 | I | V | V | – | – | – | – | – | – |
| 53 | F | Y | – | – | – | – | – | L | – |
| 54 | I | S | – | V | – | V | – | V | – |
| 55 | K | – | – | R | – | – | – | – | – |
| 57 | R | – | – | – | – | – | K | – | – |
| 58 | Q | – | E | – | – | – | – | – | – |
| 60 | D | – | – | – | E | – | – | – | – |
| 61 | Q | – | – | – | – | – | Y | – | – |
| 62 | I | V | – | – | V | V | – | – | – |
| 63 | L | Q | P | P | P | P | – | P | P |
| 64 | I | – | – | V | – | – | – | – | – |
| 66 | I | – | – | V | – | – | – | – | – |
| 67 | C | – | – | – | – | F | – | – | – |

*Appendix 1—table 3 continued on next page*

*Appendix 1—table 3 continued*

| Position | Consensus | I50V ME | I50V LE | I84V ME | I84V LE | L90M ME | L90M LE | V82A ME | V82A LE |
|---|---|---|---|---|---|---|---|---|---|
| 71 | A | V | T | – | – | V | – | V | – |
| 72 | I | V | V | – | – | K | – | – | – |
| 73 | G | – | – | T | – | S | – | – | – |
| 74 | T | S | – | – | – | – | – | – | – |
| 76 | L | – | – | – | – | – | – | – | V |
| 77 | V | I | – | – | – | – | I | – | – |
| 79 | P | – | – | A | – | A | – | – | – |
| 82 | V | **A** | **I** | – | – | – | **I** | **A** | **A** |
| 84 | I | – | – | **V** | **V** | **V** | – | – | **V** |
| 88 | N | – | – | – | D | – | – | – | – |
| 89 | L | – | V | – | – | – | – | – | I |
| 90 | L | – | – | **M** | **M** | **M** | **M** | – | – |
| 93 | I | L | L | L | – | – | – | – | – |
| 95 | C | – | – | F | – | – | – | – | – |
| HD | | 15 | 15 | 13 | 13 | 14 | 14 | 9 | 9 |

Residues same as consensus are shown as '–', mutations are shown as the one letter abbreviated alphabet encoding the mutant residue (in bold are primary drug-resistance mutations appearing at more than 1% frequency).

DOI: https://doi.org/10.7554/eLife.50524.023

