## [Decision Letter]

[Editors’ note: a previous version of this study was rejected after peer review, but the authors submitted for reconsideration. The first decision letter after peer review is shown below.]

Thank you for submitting your work entitled "Epistasis and entrenchment of drug resistance in HIV-1 subtype B" for consideration by *eLife*. Your article has been reviewed by a Senior Editor, a Reviewing Editor, and two reviewers. The reviewers have opted to remain anonymous.

Our decision has been reached after consultation between the reviewers. Based on these discussions and the individual reviews below, we regret to inform you that your work will not be considered further for publication in *eLife*.

The topic of your paper is very important, and indeed epistasis and the effects of coupled mutations in HIV proteins on the fitness of HIV is documented. However, as the reviews below note, because your study is based on individuals undergoing treatment, it is impossible to separate correlated selection pressures from epistatic effects.

Reviewer #1:

Here Biswas and collaborators apply coevolutionary analysis methods to study the entrenchment of HIV drug resistance mutations in the protease, integrase, and reverse transcriptase proteins. Entrenchment refers to an increasing preference for the mutant (i.e., resistant) residue as epistatic accessory mutations are accumulated. The authors demonstrate an impressive ability to predict the frequency of resistance mutations in different genetic backgrounds. They then characterize how entrenchment depends on the number of accumulated resistance-associated mutations, and which specific mutations are most strongly associated with entrenchment.

Specific comments:

1) One might reasonably hypothesize that the reason that drug resistance mutations are more likely to appear together in the Potts model is because the model is capturing correlated selection pressures rather than true epistasis. In other words, in patients who are treated by one drug, there is selection for the accumulation of resistance mutations for that drug. No such selection is present in patients not treated by that drug. Thus, whether the resistance/accessory mutations are truly epistatic or not, they will tend to appear either clustered together in sequences or not at all. This could then be reflected as a network of positive epistatic couplings in the Potts model. Butler et al., cited by the authors, used sequence data obtained before the widespread use of protease inhibitors to infer an Ising model that contained positive epistatic couplings between resistance sites. In principle, this approach should limit the effect of correlated selection pressures due to differences in drug treatment among different individuals. Butler and collaborators further found some external evidence for true epistasis between a resistance mutation at site 84 in protease and other accessory mutations. In the present work, given that the MSA contains sequences from individuals who are drug-experienced, are there independent lines of evidence that support the hypothesis of true epistatic interactions between associated resistance mutations? And more broadly, how would one deal with the confounding influence of correlated selection pressures?

2) As a possible suggestion, one might expect that inferring Potts parameters using the pseudolikelihood method would provide an even better fit of the probability of different mutations at a given site as a function of the sequence background, since this is precisely what the pseudolikelihood algorithm is optimizing. In contrast, the canonical maximum entropy approach must simultaneously fit all low-order correlations.

3) Some of the figures and tables are challenging to interpret. For example, in Supplementary file 1 both the most (ME) and least (LE) residues at position 17 for the L74M mutation are N. This result also appears for other mutations and positions in other tables. This needs further explanation. The plot in the inset of Figure 1B is also a bit challenging to interpret at first, and the small size of the numbers/labels in the inset plot makes it difficult to read at standard resolution. The ordering of resistance mutations in Figure 1D is confusing. The linear order would suggest order along the sequence, but mutations within each sequence are presented in reverse order along the protein sequence. Integrase is also placed before reverse transcriptase. This figure would likely be more clear if the same color scheme was maintained, but the mutations were reordered according to their positions in Pol.

Reviewer #2:

The study analyzes molecular co-evolution in sequences of HIV from drug-exposed patients, applying a Potts model to infer fitness effects of mutations and their dependence on the genetic background. The authors use the inferred model to analyze the notion of entrenchment - that a mutation tends to become more difficult to revert, over time, due the epistatic interactions with subsequent substitutions in the genetic background on which it first arose. The authors report this trend in three different HIV proteins that are targets of various drug treatments.

The intellectual basis of this study is sound and interesting - that epistasis can shape and channel molecular evolution - and it builds on a broader theoretical literature that has demonstrated such phenomenon for protein evolution in general (e.g. Pollack et al., 2012; Shah et al., 2015). Demonstrating this effect for real-world sampled protein sequences, in the setting of HIV protein evolution in response to drugs, is an especially nice project - as it would provide real-world relevance, in a clinical setting, for a broad body of theoretical work.

Despite the enthusiasm for the topics, I am disappointed by a large number of inconsistencies in the reported results - which do not, in the end, seem to demonstrate the type of entrenchment the authors start out to explore. And I'm also disappointed by the presentation of the model and analysis, which is ambiguous or unclear in both main text and Materials and methods section. I am left without a clear understanding of what the authors claim to have demonstrated, or what specific model inferences they are reporting in most of their figures.

These two problems - whether their analysis actually addresses the established concept of entrenchment, and whether the writing/notation is clear enough to explain what analyses they actually performed – occur throughout the manuscript, and so I will address instances of these two problems in a linear fashion throughout the work:

1) The Potts model appears to be massively overfit. Even after artificially reducing the alphabet size from 20 acids to 4, the authors seem to be fitting roughly 380,000 parameters to account for all pairwise interactions between sites. And yet, as far as a I can tell, the authors have only ~20,000 sequences. So, they are fitting an order of magnitude more parameters than they have data. Even so, they make a big deal of the fact the model makes good "predictions" for the frequency of sequences observed (Figure 1B)- but it is not clear if these prediction were made with cross-validation (e.g. fitting on one part of the data, and prediction for another part), and little discussion of overfitting is provided, if any. In any case, the notion that pairwise maxent Potts-like models can do a decent job of explaining sequence variability is not a new result and has been shown in dozens of papers.

1b) The authors discuss their Potts model as providing a general probability for the observing a mutation alpha at site i in a genetic background of sequence S. But, in fact, the model depends only on pairwise interactions, the full sequence S. The authors should clarify this in the main text.

1c) It is not clear what reference sequence is used as the "wildtype" S throughout the analysis. Is this a consensus sequence from the database? This should be explained in the main text.

2) The main results on entrenchment (Figure 2) are not described precisely and seem to contradict the notion of entrenchment to which the authors refer in their introduction. Entrenchment according to Pollack et al., occurs when a substitution is initially neutral or slightly beneficial (so that its reversion is neutral or only slightly deleterious), but when subsequent substitutions in the protein render the focal mutation highly deleterious to revert. And yet the mutations discussed here are drug resistance mutations arising in the presence of drugs - and so the primary mutation must be net highly beneficial, even if it partly disrupts HIV protein function. (Indeed Figure 4 shows that the primary DRMS are favored.) And so, prima facie, this situation is quite different to Pollack at all - because the reversion is immediately disfavored, even without changes to the genetic background. Presumably the authors intend to show that the DRMs become *even more deleterious* to revert as subsequent substitutions accrue, although they do not say this in the main text and repeatedly refer to the DRMs as carrying fitness costs as the time of their emergence.

2b) Figure 2A, showing the main entrenchment results according to the Potts model, seems to contradict the notion the primary mutation is a resistance site that is beneficial at the time of its substitution (as in Figure 4). The figure reports a nearly 100-fold preference to revert the primary DRM L90M substitution in the background in which it arose - i.e. the mutation is inferred to carry a huge fitness cost (and its reversion a benefit) at the time of its origin. This does not make sense to me, as a reader, as the DRM (even if it contains some costs for protein function) must be net adaptive at the time of its substitution in a patient. Why does Figure 2A show that the drug resistance site is strongly disfavored in the genetic background in which it initially arose?

2c) The authors should define in terms of the ΔE notation when, precisely, is being plotted on the y-axis of Figure 4A.

2d) The authors state "entrenchment of key resistance mutations is likely to play an important role in emergence of drug-resistance viral strains". I cannot understand what the authors mean by this. Certainly, the initial *emergence* of a drug resistance strain depends on the adaptive benefit of DRM on the background in which it arises - and has nothing to do with future changes to the genetic background. Perhaps the authors mean that the "persistence" of the DRM over subsequent evolution depends on entrenchment?

3) [Presentation issue] The authors discuss the stabilization or destabilization of mutations in their Introduction and throughout (Table 1). I believe by "stabilization" they mean epistasis with subsequent substitutions that render the primary mutation costly to revert. But this is a terrible choice of words, because the large literature on entrenchment (Pollack, Shah etc.) has been developed for models of thermostability - and, indeed, entrenchment has been observed to arise from interactions between sites for protein stability. The authors do not seem to be using the word "stability" here to mean thermostability, which is tremendously confusing when referring to a literature that is all about the effects of mutations, and their interactions, on thermostability.

4) The authors make a big deal of their analysis of "entrenchment of DRMs" in the population - meaning that drug resistance mutations are beneficial in over 50% of the sequences sampled/fit. I do not find this surprising or even attributable to entrenchment at all - because, after all, the sequences being used to fit the model are sequences from drug-experienced HIV populations, where the drug resistance mutation will be net beneficial *at the time of its original occurrence* and even without any further substitutions. This is shown directly in Figure 4, for example.

5) When the authors report that they have "verification" of entrenchment I assumed they meant they had direct data on fitness measurements for a mutation in one genetic background, and in a subsequent background. But in fact, the "verification" is simply relative to the fitted Potts model. By contrast, when the authors do compare the model to actual measurements of fitness effects (subsection “Molecular clones and the effects of specific backgrounds”), the authors seem to be saying that the predictions of the Potts model do not match the empirical measurements, and they attribute this to limitation of the mutagenesis experiments, as opposed to a deficiency of the Potts model. Comparison to data would have been the only thing that really convinces me that any statistical effects of their model are "verified".

5b) Also, I had a hard time understanding what conclusions the authors draw from Figure 5, or even what the axes really denote in Figure 5. The notation Δ E_patient does not seem to be defined in the main text. The authors seem to be showing effects of mutations predicted by the Potts model fitted to patient data versus their predicted effects in the specific genetic backgrounds used for mutagenesis experiments - but it's not clear if that's what they're plotting, in fact, or what we are supposed to conclude from Figure 5.

---

## [Author Response]

[Editors' note: the author responses to the re-review follow.]

Reviewer #1:Here Biswas and collaborators apply coevolutionary analysis methods to study the entrenchment of HIV drug resistance mutations in the protease, integrase, and reverse transcriptase proteins. Entrenchment refers to an increasing preference for the mutant (i.e., resistant) residue as epistatic accessory mutations are accumulated. The authors demonstrate an impressive ability to predict the frequency of resistance mutations in different genetic backgrounds. They then characterize how entrenchment depends on the number of accumulated resistance-associated mutations, and which specific mutations are most strongly associated with entrenchment.Specific comments:1) One might reasonably hypothesize that the reason that drug resistance mutations are more likely to appear together in the Potts model is because the model is capturing correlated selection pressures rather than true epistasis. In other words, in patients who are treated by one drug, there is selection for the accumulation of resistance mutations for that drug. No such selection is present in patients not treated by that drug. Thus, whether the resistance/accessory mutations are truly epistatic or not, they will tend to appear either clustered together in sequences or not at all. This could then be reflected as a network of positive epistatic couplings in the Potts model. Butler et al., cited by the authors, used sequence data obtained before the widespread use of protease inhibitors to infer an Ising model that contained positive epistatic couplings between resistance sites. In principle, this approach should limit the effect of correlated selection pressures due to differences in drug treatment among different individuals. Butler and collaborators further found some external evidence for true epistasis between a resistance mutation at site 84 in protease and other accessory mutations. In the present work, given that the MSA contains sequences from individuals who are drug-experienced, are there independent lines of evidence that support the hypothesis of true epistatic interactions between associated resistance mutations? And more broadly, how would one deal with the confounding influence of correlated selection pressures?

The reviewer’s comments suggest that we need to address three important questions: (1) Can we distinguish between correlations induced by “true epistastic interactions” and those due to correlated selection pressure induced by drugs; (2) Do our conclusions about the entrenchment of drug resistance mutations depend on whether the origins of the mutational correlation patterns can be separated into “intrinsic epistasis” and effects due to correlated selection pressure induced by drugs; (3) Is the Potts model we build from the drug experienced dataset contaminated by artifacts associated with different patients being given different drug regimens.

1) Can we distinguish between correlations induced by “true” or “intrinsic” epistasis and those induced by correlated selection pressure induced by drugs?

To begin with, we note that while reviewer #1 refers to epistatic effects which are independent of drug selection pressure as “true epistasis”, we prefer to describe them as “intrinsic epistasis”, since correlated mutations induced by drug selection pressure may also be considered to be a form of epistasis. We use the term “effective epistasis” to refer to mutational correlations induced by the combined effects of “intrinsic epistasis” and correlated selection pressure due to drugs. We have added a new subsection “Effective epistasis in the presence of drug selection pressure” where we discuss the distinction between “intrinsic epistasis” and correlated selection pressure.

In the case of immune response selection pressure Shekhar et al., 2013 suggest that due to the diversity of host immune responses among the HIV population, a Potts model fit to sequences from many different hosts can effectively capture the “intrinsic” fitness landscape of the virus because the selective effects of immune pressure are averaged in a way that can be modeled by adjusting the fields of the Potts model, leaving the couplings unchanged.

We have available to us from the Stanford HIV database, a drug experienced sequence data set and a drug naïve dataset. We have fit a Potts model for HIV-1 Subtype B Protease to each dataset individually. Comparisons of the Potts statistical energy values ΔE predicted by the drug naïve and drug experienced models for mutating from the wild type to the drug resistance mutations for four HIV PR DRMS as a function of the background sequence are presented in subsection “Effective epistasis in the presence of drug selection pressure” of the revised manuscript. There is a high correlation (r > 0.8) between the probabilities of observing the drug resistance mutation relative to the wild type in a given background, when the Potts model is parameterized on the drug naïve dataset as compared with the drug experienced dataset. This means that “intrinsic” epistatic effects have a large influence on the virus evolving under drug selection pressure.

It is possible to carry out a more quantitative analysis of how the application of drug selection pressure changes the intrinsic mutational correlation patterns and probabilities of observing DRMs in specific sequence backgrounds using multi-canonical reweighting techniques familiar to researchers who work with Ising or Potts models. While such an analysis is beyond the scope of the present work, in a future communication we will provide a detailed comparison of the drug naïve and drug experienced Potts models and the relationship between them, as such an analysis can provide insights into sequence patterns which are most strongly associated with the transmission of drug resistance.

2) Do our conclusions about the entrenchment of drug resistance mutations depend on whether the origins of the mutational correlation patterns can be separated into “intrinsic epistasis” and effects due to correlated selection pressure induced by drugs?

As far as the goal of the current work is concerned, to demonstrate that the Potts model can accurately predict the effect of the sequence background on the probability of observing a resistance mutation in the drug experienced population, and therefore identify those sequences which are highly favoring (entrenching) or highly disfavoring the mutation, it is not necessary to separate the effects of intrinsic epistasis from those due to correlated selection pressure induced by drugs. The data we are comparing the predictions of our Potts models to are the observed frequencies (prevalences) in the Stanford HIV drug experienced sequence database of drug resistance mutations, in sets of sequence backgrounds classified by our model as either favoring or disfavoring the DRM. These are non-trivial predictions; the observations confirm the predictions. Put another way, the Potts model parameterized on the drug experienced dataset is a powerful classifier that can identify complex sequence patterns that highly favor (entrench) or disfavor each drug resistant mutant. It should be noted that the frequency of any individual drug resistance mutation (DRM) appearing in the drug experienced Stanford HIV database is typically small. See our response to reviewer #2 comment 4) for additional discussion of the distinction between the frequency of a DRM appearing in the drug experienced population and the “entrenchment” of that DRM.

3) Is the Potts model we build from the drug experienced dataset contaminated by an artifact associated with different patients being given different drug regimens? This could result in spurious apparent correlations predicted by the Potts model.

Reviewer #1 also expresses a concern that if the selection pressures induced by different drugs are significantly different, then this could lead to artifacts when parameterizing a Potts model on a drug experienced dataset, as the model would potentially be sensitive to the details of how many patients in the dataset are exposed to each drug. While we cannot definitively rule out such an artifact, two lines of evidence suggest that if such effects are present, they are small.

First, we note that HIV drugs are frequently given in combination, and that many of the drugs have similar resistance patterns. We include a comparison of these patterns for different Protease inhibitors in Figure 8 of the revised manuscript. Secondly, as noted above there is a strong correlation between the ΔE values predicted using the Potts model parameterized on the drug naïve dataset and the one parameterized on the drug experienced set. If the details concerning the number of individuals which were given specific drugs and combination therapies strongly influenced the Potts model parameterized on the drug experienced dataset, we would not expect the “effective epistatic” effects on ΔE inferred by the drug experienced model to be highly correlated with the “intrinsic epistatic” effects of the drug naïve model. These observations mitigate possible artifacts associated with the fact that individual HIV patients represented in the drug experienced dataset may be undergoing different drug therapies. We added a statement about in subsection “Effective epistasis in the presence of drug selection pressure” of the revised manuscript.

2) As a possible suggestion, one might expect that inferring Potts parameters using the pseudolikelihood method would provide an even better fit of the probability of different mutations at a given site as a function of the sequence background, since this is precisely what the pseudolikelihood algorithm is optimizing. In contrast, the canonical maximum entropy approach must simultaneously fit all low-order correlations.

We are using an approach which optimizes the maximum entropy Boltzmann likelihood function by fitting the bivariate marginals of the Potts model to the observed bivariate marginals using MCMC sampling. This is a more compute intensive approach to inferring the parameters of the Potts Hamiltonian, but it is also more accurate as the errors in the bivariate and univariate marginals are much smaller than those obtained by the pseudolikelihood method. This has been demonstrated in the literature; for instance, the supplementary information in Barton et al., 2016 shows large relative errors in the generated univariate marginals and the pairwise correlations using the pseudolikelihood method, and in Jacquin et al., 2016, Barton et al., 2016, it is shown that the pseudolikelihood method is not "generative", meaning sequences generated using it have modified bivariate marginals. The agreement we obtain using our MCMC GPU implementation of inverse inference, between predicted and observed frequencies of drug resistance mutations in different sets of backgrounds is excellent, and probably cannot be matched using more approximate inverse inference methods like the pseudo-likelihood.

3) Some of the figures and tables are challenging to interpret. For example, in Supplementary file 1 both the most (ME) and least (LE) residues at position 17 for the L74M mutation are N. This result also appears for other mutations and positions in other tables. This needs further explanation. The plot in the inset of Figure 1B is also a bit challenging to interpret at first, and the small size of the numbers/labels in the inset plot makes it difficult to read at standard resolution. The ordering of resistance mutations in Figure 1D is confusing. The linear order would suggest order along the sequence, but mutations within each sequence are presented in reverse order along the protein sequence. Integrase is also placed before reverse transcriptase. This figure would likely be more clear if the same color scheme was maintained, but the mutations were reordered according to their positions in Pol.

Supplementary file 1 in the updated supplement presents a pair of sequences for each mutation; one being the most entrenching for that mutation and the other the least entrenching (the probability of observing the mutation relative to the wildtype at this position is the smallest in the least entrenching background). There will be positions at which the same accessory mutation appears in both the most and least entrenching backgrounds; for example, the mutation 17N, which is accessory to L74M. The background sequences which are the most and least entrenching for any resistance mutation is a collective property of the set of all mutations in the corresponding background; any particular accessory mutation may appear in both backgrounds.

As per the reviewer’s suggestion, we have changed the ordering of mutations in Figure 1D according to their positions in *Pol* and resized Figure 1 for clarity.

Reviewer #2:The study analyzes molecular co-evolution in sequences of HIV from drug-exposed patients, applying a Potts model to infer fitness effects of mutations and their dependence on the genetic background. The authors use the inferred model to analyze the notion of entrenchment – that a mutation tends to become more difficult to revert, over time, due the epistatic interactions with subsequent substitutions in the genetic background on which it first arose. The authors report this trend in three different HIV proteins that are targets of various drug treatments.The intellectual basis of this study is sound and interesting – that epistasis can shape and channel molecular evolution – and it builds on a broader theoretical literature that has demonstrated such phenomenon for protein evolution in general (e.g. Pollack et al., 2012; Shah et al., 2015). Demonstrating this effect for real-world sampled protein sequences, in the setting of HIV protein evolution in response to drugs, is an especially nice project – as it would provide real-world relevance, in a clinical setting, for a broad body of theoretical work.

We appreciate the reviewer’s enthusiasm for the topic of the paper.

Despite the enthusiasm for the topics, I am disappointed by a large number of inconsistencies in the reported results – which do not, in the end, seem to demonstrate the type of entrenchment the authors start out to explore. And I'm also disappointed by the presentation of the model and analysis, which is ambiguous or unclear in both main text and Materials and methods section. I am left without a clear understanding of what the authors claim to have demonstrated, or what specific model inferences they are reporting in most of their figures.These two problems – whether their analysis actually addresses the established concept of entrenchment, and whether the writing/notation is clear enough to explain what analyses they actually performed – occur throughout the manuscript, and so I will address instances of these two problems in a linear fashion throughout the work:

We have made revisions to the manuscript to make it clearer following the suggestions of the reviewers, and in light of what appears to be a serious misunderstanding of our work at several places by reviewer #2. We do not believe that there are inconsistencies in the reported results; it is apparent from the reviewer’s comments that we have done an inadequate job of describing our work. Several statements made by referee #2 are incorrect. We have revised the manuscript to address the issues and concerns raised by referee #2 as described below.

1) The Potts model appears to be massively overfit. Even after artificially reducing the alphabet size from 20 acids to 4, the authors seem to be fitting roughly 380,000 parameters to account for all pairwise interactions between sites. And yet, as far as a I can tell, the authors have only ~20,000 sequences. So, they are fitting an order of magnitude more parameters than they have data. Even so, they make a big deal of the fact the model makes good "predictions" for the frequency of sequences observed (Figure 1B) - but it is not clear if these prediction were made with cross-validation (e.g. fitting on one part of the data, and prediction for another part), and little discussion of overfitting is provided, if any. In any case, the notion that pairwise maxent Potts-like models can do a decent job of explaining sequence variability is not a new result and has been shown in dozens of papers.

Overfitting is an important issue and we appreciate the suggestion that we should discuss it more thoroughly. We have recently published a detailed analysis of overfitting effects in the context of inverse inference of Potts model parameters from protein multiple-sequence alignments (Haldane and Levy, 2019). We have added a discussion of overfitting effects to the main text in subsection “Statistical Robustness of HIV Potts models” which we summarize here.

The reviewer suggests that we are overfitting our model because the number of model parameters (θ) is much larger than the number of sequences in our dataset (N). However, we explain in Haldane and Levy, 2019 why this is not the correct comparison to make. First, our model is not fit directly to the sequences but rather to the bivariate marginals of the MSA, and there are an equal number of model parameters θ as there are bivariate marginals. The inference problem is therefore neither overconstrained nor underconstrained. In addition, while for (linear) regression overfitting occurs when the number of datapoints is less than the number of model parameters, this is not the case for inference methods in general.

In Haldane and Levy, 2019 we explain how overfitting ultimately arises due to finite-sampling error in the bivariate marginal frequencies used as input to the inference procedure, which are computed from the MSA of N sequences. Each bivariate marginal f is estimated from a sample of size N, and its statistical error is the multinomial mean-square error f(1-f)/N. This statistical error in the bivariate marginals leads to error in the inferred model parameters.

Based on this fact, in Haldane and Levy, 2019 we go on to derive the dependence of model quality on the number of parameters and the number of sequences in the MSA, and we show how one can fit accurate Potts models to MSAs containing only thousands of effective sequences with negligible overfitting, even though there are many more Potts model parameters than sequences. We illustrate this using a simple-to-understand toy model and demonstrate it numerically for realistic Potts models. We find that the degree of overfitting can be quantified using the "signal-to-noise ratio" (SNR), which is a function of the number of model parameters θ, the degree of conservation of the MSA as measured by a value called Χ^2^, and the number of sequences N. While the SNR is related to the ratio of number of parameters θ to number of sequences N, it contains other terms including Χ^2.^. In order to demonstrate that our Potts models for HIV proteins are not overfit, we have added a discussion of the SNR of our models to the revised manuscript showing that they have suitable SNR and satisfy the criteria described in Haldane and Levy, 2019 for estimating statistical energy differences without overfitting. Our paper (Haldane and Levy, 2019) also discusses cross-validation of Potts statistical energies and quantities which can be derived from the statistical energies such as ΔE.

We think reviewer #2’s comment “In any case, the notion that pairwise maxent Potts-like models can do a decent job of explaining sequence variability is not a new result and has been shown in dozens of papers.” is inappropriate. We have been active and publishing in this field since 2012 and are well versed in the literature which is extensively cited. Our analysis of the effects of the sequence background on the likelihood of drug resistance mutations in HIV-1, which first appeared in 2017 (Flynn et al., 2017) and which is presented in a more complete form in the current manuscript where the predictions are verified, is highly novel, there is nothing comparable in the literature. Our analysis of the entrenchment of DRMS for HIV-1 is not simply a calculation of the fitness of single point mutations, which has been reported by several groups working with Potts models of sequence variability. In order to demonstrate the entrenchment of a DRM by the background it is necessary to be able to capture the effects of many simultaneous mutations on DRM sites and how those effects change as the sequence background changes. This is related to our ability to model higher order marginals of the distribution, which is possible using our MCMC method to infer the Potts model. It would be quite difficult to obtain results concerning the entrenchment of DRMS of comparable quality using more approximate inference methods.

1b) The authors discuss their Potts model as providing a general probability for the observing a mutation alpha at site i in a genetic background of sequence S. But, in fact, the model depends only on pairwise interactions, the full sequence S. The authors should clarify this in the main text.

While the Potts Hamiltonian model contains only pairwise interaction terms, the Potts effective potential function induces higher order correlations and the probability of a given sequence cannot be decomposed into a sum of products of lower order terms. In other words, pairs of residues interact with each other both directly (through a direct coupling in the Hamiltonian) and indirectly through chains of interactions involving one or more intermediate residues. As a consequence, the higher order marginals of the distribution cannot be expressed as an explicit function of the pair correlations. Our ability to predict the effect of the background on the likelihood of a drug resistance mutation is related to the ability of our MCMC algorithm to infer a Potts model of sequence co-variation which is sufficiently accurate to infer higher order marginals of the distribution as described in Flynn et al., 2017; Haldane et al., 2018. The Potts model Hamiltonian with pairwise terms is both necessary and sufficient to capture many observable effects that involve higher order correlations, like the effect of the sequence background on the likelihood of a DRM. Whether or not Hamiltonian functions which include triplet or even higher order terms are needed to capture other properties of the HIV sequence ensembles is a question that is not strictly germane to our paper, and in any case is best studied first in toy models because of the data limitations inherent in HIV sequence databases. We have added a discussion of this point to the main text in subsection “Epistasis: the effect of the background on primary resistance mutations in HIV” and subsection “Model inference”.

1c) It is not clear what reference sequence is used as the "wildtype" S throughout the analysis. Is this a consensus sequence from the database? This should be explained in the main text.

For all three proteins (PR, RT, IN) the reference sequence referred to as “wildtype” is the consensus subtype B sequence, as obtained from an alignment of subtype B sequences maintained in the Los Alamos HIV sequence database, which is frequently used as the reference for mutation studies. Where the consensus sequence can be obtained is stated the revised Materials and methods section. Since this reference sequence is often referred to as the “consensus wild-type subtype B” sequence, we have adopted this terminology in the main text and in the Materials and methods section.

2) The main results on entrenchment (Figure 2) are not described precisely and seem to contradict the notion of entrenchment to which the authors refer in their introduction. Entrenchment according to Pollack et al. occurs when a substitution is initially neutral or slightly beneficial (so that its reversion is neutral or only slightly deleterious), but when subsequent substitutions in the protein render the focal mutation highly deleterious to revert. And yet the mutations discussed here are drug resistance mutations arising in the presence of drugs – and so the primary mutation must be net highly beneficial, even if it partly disrupts HIV protein function. (Indeed Figure 4 shows that the primary DRMS are favored.) And so, prima facie, this situation is quite different to Pollack at all – because the reversion is immediately disfavored, even without changes to the genetic background. Presumably the authors intend to show that the DRMs become *even more deleterious* to revert as subsequent substitutions accrue, although they do not say this in the main text and repeatedly refer to the DRMs as carrying fitness costs as the time of their emergence.2b) Figure 2A, showing the main entrenchment results according to the Potts model, seems to contradict the notion the primary mutation is a resistance site that is beneficial at the time of its substitution (as in Figure 4). The figure reports a nearly 100-fold preference to revert the primary DRM L90M substitution in the background in which it arose – i.e. the mutation is inferred to carry a huge fitness cost (and its reversion a benefit) at the time of its origin. This does not make sense to me, as a reader, as the DRM (even if it contains some costs for protein function) must be net adaptive at the time of its substitution in a patient. Why does Figure 2A show that the drug resistance site is strongly disfavored in the genetic background in which it initially arose?

We use the term entrenchment to refer to the fact that the sequence background has a large effect on the relative likelihood of observing a drug resistance mutation vs. a consensus wildtype residue at a DRM site. For some sequence backgrounds with several accessory mutations the background is highly favorable for the DRM; we refer to these mutation patterns as “entrenching”, while for other backgrounds with the same total number of associated mutations, those backgrounds highly disfavor the DRM. Our Potts Hamiltonian model is able to predict these sequence specific effects (entrenching vs. disfavoring DRMS) with high fidelity, even for sets of sequences conditional on having the same total Hamming distance from the wild type consensus sequence. This is an achievement that would not be possible using more approximate inverse inference techniques to construct the Potts Hamiltonian.

Perhaps the reviewer’s questions 2 and 2b are due to a misunderstanding concerning the information contained in Figure 2? It appears that the reviewer is interpreting Figure 2 as a time ordering of the appearance of the L90M mutation in backgrounds with increasing numbers of mutations. There is no information about time ordering of the appearance of L90M in Figure 2. We are not currently using the Potts model to infer information about the kinetics by which drug resistance is acquired. In fact, the drug resistance mutation 90M appears in the Stanford drug experienced dataset only 2 times in the wild type consensus background, while the wild-type consensus residue L90 appears 40 times in the wild type consensus background in the same dataset. Even though L90M is a drug resistance mutation, it is deleterious in the consensus wildtype background because of intrinsic epistatic effects, and according to the Stanford drug experienced database, the L90M mutation in the consensus background is ~ 20x less likely to be observed than the wild-type residue (L90) in the same consensus background sequence.

While Figure 2 should not be thought of as a time ordering of the appearance of L90M in a population, we can speculate about kinetics given the information in Figure 2. A possible scenario would be that L90M first appears in backgrounds where the L90M mutation is almost neutral, there are many such sequences with a Hamming distance of ~7 to 10 mutations from the consensus wildtype. Many of these backgrounds are almost neutral for L90M. Additional mutations, sometimes accompanied by reversions, will in general increase the likelihood of L90M leading to stronger entrenchment. Such a scenario is consistent in a general way with the time dependent model of entrenchment described by Shah et al., (2015) and Pollock et al., (2012), even though our current model does not include kinetics. But there are differences with the scenario envisioned by these authors in that the initial HIV-1 DRM mutation in the wildtype consensus sequence need not be almost neutral, since the path to entrenchment of drug resistance mutations need not pass through that sequence containing 1 DRM in the consensus background (the paper by Shah et al., does describe a similar scenario). We have revised the main text (subsection ““Entrenchment” of a primary resistance mutation and its verification”, Discussion section) to make it clear that our model is not a kinetic model, and that the L90M mutation is only rarely observed in the wild type consensus background in the drug experienced Stanford database. We plan to investigate the time evolution of the appearance of drug resistance in specific backgrounds using explicitly kinetic models in upcoming work.

2c) The authors should define in terms of the ΔE notation when, precisely, is being plotted on the y-axis of Figure 4A.

It is not clear whether the reviewer is referring to Figure 2A or Figure 4? Concerning the Y-axis on the rhs of Figure 2A, D,E is the difference in the Potts statistical energy of the sequence with the consensus wild type residue L at position 90 minus the Potts statistical energy of the drug resistance mutation M at position 90. A positive value ΔE means the mutant is favored. ΔE is the log of the probability of observing the mutant M in the background relative to the wild type residue in the background. The box plots show the statistics (median, and quartiles) for ΔE.

Figure 4 shows the distributions of ΔE values for key residues associated with drug resistance in HIV-1 IN. The green histogram shows the calculated ΔE for every drug resistance position in sequences where the drug resistance mutation is present, while the blue histogram shows the calculated ΔE to all other residue types in the same sequence background. The relative displacement of the green from the blue distributions shows that in sequences which contain the drug resistance mutation, the mutation is predicted on average to be more likely than the consensus residue type in that background (green distribution), and mutations to other residue types in that same sequence background are less likely (blue distribution). The green histogram is normalized to the total drug resistance mutation count in the Stanford HIV database (the total number of drug resistance mutations in each sequence times the total number of sequences), while the blue histogram is normalized to the total number of other possible mutations excluding the drug resistance mutation. This is made clear in the main text in subsection “Fitness and degree of entrenchment of observed resistance mutations: why some mutations are seen and others are not” and the legend for Figure 4.

2d) The authors state "entrenchment of key resistance mutations is likely to play an important role in emergence of drug-resistance viral strains". I cannot understand what the authors mean by this. Certainly, the initial *emergence* of a drug resistance strain depends on the adaptive benefit of DRM on the background in which it arises – and has nothing to do with future changes to the genetic background. Perhaps the authors mean that the "persistence" of the DRM over subsequent evolution depends on entrenchment?

We can understand why a reader might be confused by the statement "entrenchment of key resistance mutations is likely to play an important role in emergence of drug-resistance viral strains". We have revised the text in the Discussion section to provide a better explanation of what we intended to say concerning the relationship between entrenchment and the emergence of drug resistance. In the new text we state clearly that in this manuscript we are not modeling the time development of drug resistance or identifying evolutionary paths by which drug resistance is acquired. However, our construction of Potts models for both the drug naïve and drug experienced populations, sets the stage for kinetic studies of the pathways by which drug resistance is acquired. In order to identify HIV sequences which are most responsible for conferring drug resistance, we need to know the probabilities of observing highly entrenching sequences in both the drug naïve and drug experienced populations. As we discuss in the Discussion section of the revised manuscript, multicanonical reweighting techniques that can be applied to Ising and Potts models have been developed for this purpose.

Concerning the reviewer’s statement “Certainly the initial *emergence* of a drug resistance strain depends on the adaptive benefit of DRM on the background in which it arises”, we emphasize again that the initial “emergence” of a drug resistance mutation does not need to appear in the consensus wild type background (see our response to reviewer #2 comment 2 and 2b above), and for example the Protease resistance mutation L90M is rarely observed in the wildtype consensus background in the Stanford drug experienced dataset.

3) [Presentation issue] The authors discuss the stabilization or destabilization of mutations in their Introduction and throughout (Table 1). I believe by "stabilization" they mean epistasis with subsequent substitutions that render the primary mutation costly to revert. But this is a terrible choice of words, because the large literature on entrenchment (Pollack, Shah etc.) has been developed for models of thermostability – and, indeed, entrenchment has been observed to arise from interactions between sites for protein stability. The authors do not seem to be using the word "stability" here to mean thermostability, which is tremendously confusing when referring to a literature that is all about the effects of mutations, and their interactions, on thermostability.

We have removed the words “stabilizing” and “destabilizing” when referring to mutations as we agree that these words are often used in the context of the effects of a mutation on “protein stability” or “thermostability”. Instead we use the terms “favoring” or “disfavoring” to describe the correlated effects of sequence backgrounds on the probability of a drug resistance mutation appearing in that background.

The Potts model infers the probabilities of sequences in a population; several papers have characterized how well Potts statistical energies track fitness. There are many different ways that mutations can affect fitness; they include mutational effects on thermostability, replicative capacity, viral infectivity, etc. All of these factors will contribute to the prevalence of the sequence in the Stanford HIV database. (Also, see our answer to reviewer #2 comment 4 next).

4) The authors make a big deal of their analysis of "entrenchment of DRMs" in the population – meaning that drug resistance mutations are beneficial in over 50% of the sequences sampled/fit. I do not find this surprising or even attributable to entrenchment at all – because, after all, the sequences being used to fit the model are sequences from drug-experienced HIV populations, where the drug resistance mutation will be net beneficial *at the time of its original occurrence* and even without any further substitutions. This is shown directly in Figure 4, for example.

Regrettably, the reviewer appears to misunderstand the points we make about “entrenchment of DRMS” and therefore misunderstand the main theme of our paper. The reviewer’s statement that “drug resistance mutations are beneficial in over 50% of the sequences sampled/fit” is incorrect. This is clear simply from an analysis of the frequencies (univariate marginals) of the four classes of drug resistance mutations in the three target proteins (RT mutations against NRTIs and NNRTIs, PR mutations against PIs, and IN mutations against INSTIs). In fact, for all four classes of DRMs, the frequency with which each DRM appears in the Stanford HIV drug experienced database is almost always under 50% (there is one exception, the NRTI resistance mutation M184V); and the frequencies of individual DRMS in the drug experienced dataset are usually much smaller than 50%.

We use the term “Entrenchment of a DRM in the population” to refer to the fact that for most DRM mutations (the exception being NNRTIs), we observe that the DRM is favored over the consensus residue type at the corresponding DRM position (i.e. the Potts energy change is ΔE >0 at that position, and this depends on the residues at every other position) for more than 50% of the sequences which contain the DRM. If a mutation is entrenched in the population it is then unlikely or difficult to revert in more than 50% of the sequences in which the DRM is present.

This information was contained in Table 2—source data 1, Table 2—source data 2, Table 2—source data 3, and Table 2—source data 4, of the original manuscript. We have revised the tables to make the point clearer and revised the main text in subsection “Comparative entrenchment of resistance mutations in protease, reverse transcriptase, and integrase” to point this out. Most sequence backgrounds are unfavorable for any specific DRM. Consider an example. For the π mutation D30N in Protease, only 8% of the sequences in the Stanford HIVDB contain this π mutation, but 66% of the sequences with the D30N mutation have sequence backgrounds which are entrenching, meaning that there is a greater than even chance of observing D30N in 66% of the sequences containing D30N (see Table 2—source data 3). Consider another example, the INSTI-selected mutation Q148H which occurs with frequency 19% in the Stanford Drug Experienced dataset, but in 98% of these sequences (containing Q148H), the Potts model predicts that the Q148H mutation is more likely than not (i.e. that it is entrenched in these sequences). We have revised the text in subsection “Comparative entrenchment of resistance mutations in protease, reverse transcriptase, and integrase” to make it very clear that there is a distinction between the frequency of observing a DRM (which is usually low), and the likelihood that the DRM is “entrenched” in the sequences which contain that mutation (which is usually high).

Entrenchment in the population is better defined with an example the second paragraph of subsection “Comparative entrenchment of resistance mutations in protease, reverse transcriptase, and integrase”. It is also discussed in the fifth paragraph of subsection “Comparative entrenchment of resistance mutations in protease, reverse transcriptase, and integrase” leading to a discussion of sequences containing at least one entrenched DRM.

5) When the authors report that they have "verification" of entrenchment I assumed they meant they had direct data on fitness measurements for a mutation in one genetic background, and in a subsequent background. But in fact, the "verification" is simply relative to the fitted Potts model. By contrast, when the authors do compare the model to actual measurements of fitness effects (subsection “Molecular clones and the effects of speci1c backgrounds”), the authors seem to be saying that the predictions of the Potts model do not match the empirical measurements, and they attribute this to limitation of the mutagenesis experiments, as opposed to a deficiency of the Potts model. Comparison to data would have been the only thing that really convinces me that any statistical effects of their model are "verified".5b) Also, I had a hard time understanding what conclusions the authors draw from Figure 5, or even what the axes really denote in Figure 5. The notation Δ E_patient does not seem to be defined in the main text. The authors seem to be showing effects of mutations predicted by the Potts model fitted to patient data versus their predicted effects in the specific genetic backgrounds used for mutagenesis experiments – but it's not clear if that's what they're plotting, in fact, or what we are supposed to conclude from Figure 5.

“Entrenchment” is a statement about the relative probability of seeing a drug resistance mutation as compared with the consensus residue type at a specific position, given the sequence background, which is determined by the coupling of that position to all other positions in the sequence. Our predictions of entrenchment using the Potts model are amply verified using the Potts model calculations of the change in the statistical energy ΔE as a classifier to predict the sequence backgrounds in which the drug resistance mutation is likely to be present (or absent), and then comparing the predictions with the observations from the Stanford HIV drug experienced dataset which are aggregated into groups according to the predicted values of ΔE. The comparison between our predictions and observations are presented in Table 1. We cannot emphasize too strongly that we do not fit a Potts model to observations about which sequence backgrounds are likely to favor or disfavor a DRM; the Potts model is only fit to the bivariate marginals of a very high dimensional distribution (the MSA).

A classifier of sequence backgrounds as “entrenching” or “disfavoring” towards DRMs where the predicted frequencies match the observed frequencies so well as shown in Table 1 is highly non-trivial and extremely unlikely to have happened by chance (p-value close to zero).

The subsection “Molecular clones and the effects of specific backgrounds” in the revised text presents a discussion of the results shown in Figure 5 and Figure 6. In Figure 5 we compare the predicted log likelihood (ΔE) of observing a drug resistance mutation in three specific sequence backgrounds – the molecular clones NL4-3, and HXB2, and the consensus sequence background – with the corresponding values calculated as an average over the drug experienced sequences in the Stanford HIV-1 dataset <ΔE_patients_>. We make the following points: (a) the log likelihood values of a drug resistance mutation appearing in the three specific sequence backgrounds NL4-3, HXB2, and consensus, correlates moderately with the average value in the drug experienced population and NL4-3 is most representative of the drug experienced population. Figure 5 shows that the log likelihood values are negative for most drug resistance mutations in the NL4-3 and HXB2 molecular clones, and indeed most drug resistance mutations are not present in these clones.

The reviewer’s claim that “By contrast, when the authors do compare the model to actual measurements of fitness effects (subsection “Molecular clones and the effects of speci1c backgrounds”), the authors seem to be saying that the predictions of the Potts model do not match the empirical measurement” is not correct, that is not what we are saying. As stated again for emphasis, our paper is focused on an analysis of how the sequence background affects the likelihood of observing a drug resistance mutation. We verify the predictions by comparison with the prevalence of drug resistance mutations observed in the Stanford database in particular backgrounds when the statistics are aggregated according to the predicted likelihood of observing the mutation in sequence backgrounds with Potts ΔE scores in different ranges. Potts statistical energy scores ΔE are sometimes used as a proxy for organism “fitness”; this can take on different meanings depending on which experiments (e.g. thermal stability, replicative capacity, etc.) are used to interrogate for “fitness” (Wargo and Kurath, 2012). We note that many key resistance mutations are predicted by the Potts model to be disfavored in the NL4-3 and HXB2 molecular clones, this is consistent with literature reports of the effects of these drug resistance mutations on experimental probes of fitness using the NL4-3 and HXB2 clones for these experiments (Hu and Kuritzkes, 2010; Abram et al., 2013).

Figure 6 shows the distributions of Potts statistical energy differences ΔE for two IN drug resistance mutations N155H and G140S calculated over the set of drug experienced sequences in the Stanford database. It can be seen that on average the statistical energies are much more favorable in sequence backgrounds where the drug mutation appears as compared with those where it does not. For comparison the statistical energy differences for the NL4-3 and HXB2 clones are also shown. The Potts statistical energy of a sequence is commonly interpreted to be proportional to fitness. Using changes in statistical energy as a proxy for changes in fitness when a DRM is substituted for a wild-type consensus residue, the results shown in Figure 6 imply that the fitness effects of the N155H and G140S IN mutations measured through mutagenesis experiments on specific molecular clones are likely to be quite different from the effects averaged over a distribution of sequence backgrounds represented in the Stanford HIVDB.